# The role of mPFC and MTL neurons in human choice under goal-conflict

Tomer Gazit[1,2,10], Tal Gonen[1,3,10], Guy Gurevitch [1,4,10], Noa Cohen[1,2], Ido Strauss[2,5], Yoav Zeevi[6,7], Hagar Yamin[1], Firas Fahoum [2,8], Talma Hendler [1,2,4,6,10✉] & Itzhak Fried[1,2,5,9,10]

Resolving approach-avoidance conflicts relies on encoding motivation outcomes and learning from past experiences. Accumulating evidence points to the role of the Medial Temporal Lobe (MTL) and Medial Prefrontal Cortex (mPFC) in these processes, but their differential contributions have not been convincingly deciphered in humans. We detect 310 neurons from mPFC and MTL from patients with epilepsy undergoing intracranial recordings and participating in a goal-conflict task where rewards and punishments could be controlled or not. mPFC neurons are more selective to punishments than rewards when controlled. However, only MTL firing following punishment is linked to a lower probability for subsequent approach behavior. mPFC response to punishment precedes a similar MTL response and affects subsequent behavior via an interaction with MTL firing. We thus propose a model where approach-avoidance conflict resolution in humans depends on outcome value tagging in mPFC neurons influencing encoding of such value in MTL to affect subsequent choice.

[1] Sagol Brain Institute Tel Aviv, Wohl Institute for Advanced Imaging, Tel Aviv Sourasky Medical Center, Tel Aviv, Israel. [2] Sackler Faculty of Medicine, Tel Aviv University, Tel Aviv, Israel. [3] Department of Neurosurgery, Tel Aviv Sourasky Medical Center, Tel Aviv, Israel. [4] School of Psychological Sciences, Faculty of Social Sciences, Tel Aviv University, Tel Aviv, Israel. [5] Functional Neurosurgery Unit, Tel Aviv Sourasky Medical Center, Tel Aviv, Israel. [6] Sagol School of Neuroscience, Tel Aviv University, Tel Aviv, Israel. [7] Department of Statistics and Operation Research, Tel Aviv University, Tel Aviv, Israel. [8] Epilepsy Unit, Department of Neurology, Tel Aviv Sourasky Medical Center, Tel Aviv, Israel. [9] Department of Neurosurgery, David Geffen School of Medicine, University of California Los Angeles, Los Angeles, CA, USA. [10] These authors contributed equally: Tomer Gazit, Tal Gonen, Guy Gurevitch, Talma Hendler, Itzhak Fried. ✉email: hendlert@gmail.com

Humans often find themselves facing a choice involving conflicting emotions. Spinoza defined such conflicting emotions as those which draw a man in different directions (Part IV of the Ethics, on Human Bondage). Indeed approach-avoidance behavioral choices are resolved by the human capacity to adapt goal-directed behaviors to the emotional value of prospective outcomes. Rewarding outcome serves to strengthen or reinforce context-behavior associations, thereby increasing the likelihood of future approach behavior[1]. Aversive outcomes, on the other hand, are encoded so as to avoid similar future punishment, thus encouraging avoidance behavior[2].

Animal studies have investigated the neural mechanism responsible for encoding the effects of various outcomes on subsequent behavior, mostly in the context of reinforcement learning. Accumulating evidence points to the striatum as an important region involved in signaling prediction errors (PEs)[3] and to the medial prefrontal cortex (mPFC)[4], and medial temporal lobe (MTL) as processing outcome values and valence[5,6]. Of particular importance is the known role of the hippocampus and the amygdala in forming, respectively, contextual and emotional associations[7] that guide future behavior in reinforcement learning procedures. However, less is known regarding the effects of outcome valence on the probability of subsequent behavior in situations of goal conflict.

Goal conflicts arise when we encounter potential gains and losses simultaneously within the same context[8]. Such conflicts are thought to be central to the generation of anxiety; a state of high arousal and negative outcome bias that often leads to disadvantageous dominance of choosing avoidance behavior[9,10]. Classical animal studies using goal-conflict paradigms such as the elevated plus maze (EPM)[11,12] have implicated the amygdala[13], hippocampus[9,14], and mPFC[15,16] as being crucial in triggering avoidance behavior in goal conflict situations. For example, in Kimura et al.[12], rats were punished with a delivery of an electrical shock as they consumed food (avoidance training). Over time, control animals increased their latency to enter the target box, while rats with hippocampal lesions presented impaired acquisition of such passive avoidance behavior. However, classical animal studies have not clearly differentiated the neural substrates involved in using information regarding the valence of outcomes (reward vs. punishment) for subsequent adaptation of approach behavior, from those that mediate the actual resolution of the goal conflict[17]. Schumacher et al.[17] showed that the ventral hippocampus (vHPC) is involved in the resolution of approach-avoidance conflict at the moment of

decision making rather than in learning about the value of outcomes for future decisions. On the other hand, further studies showed that the hippocampus, as well as the amygdala, seems to support learning from outcomes and thus affect future behavior. For example, Davidow et al.[5] showed that adolescents were better than adults at learning from outcomes to adapt subsequent decisions, and that this was related to heightened PE-related BOLD activity in the hippocampus. Using lesions to macaque amygdalae, Costa et al.[18] present evidence that the amygdala plays an important role in learning from outcomes to influence subsequent choice behavior. With relation to psychopathology, it has been suggested that patients suffering from depression are unable to exploit affective information to guide behavior[19]. For example, Kumar et al.[20] found reduced reward learning signals in the hippocampus and anterior cingulate in patients suffering from major depression. Disruption of prediction-outcome associations in the bilateral amygdala–hippocampal complex was found in patients with schizophrenia[21]. Yet, it remains to be seen whether these results, pointing to the significance of the MTL in the processing of outcomes and adapting behavior, are relevant to outcomes that appear in the context of an approach-avoidance conflict.

To investigate these processes, we use a rare opportunity to perform intracranial recordings from multiple sites in the MTL and mPFC of 14 patients with epilepsy (Table 1). We apply a previously validated game-like computerized task[22] that enables the measurement of goal-directed behavior (the tendency to approach) under high or low goal conflict and evaluate the neural response to the outcome of this behavior (reward or punishment, Fig. 1). During the game participants control the movement of a cartoon avatar across the screen in order to approach and capture falling coins (Reward, marked by a new Israeli shekel sign) while attempting to avoid being hit by balls that fall simultaneously (Punishment). The simultaneous appearance of these potentially rewarding and punishing cues introduces a goal-conflict behavioral decision of either approaching the reward or forfeiting it to minimize the risk of consequent punishment. To account for behavioral choice effects, the game also includes events in which outcomes occur independently of behavior (Uncontrolled condition). In this condition participants receive rewarding coins or were hit by punishing balls, regardless of their management of the avatar's movement on the screen. Reward trials during the Controlled condition are classified as either high goal conflict (HGC; more than one ball between the avatar and the reward cue) or low goal conflict (LGC; zero or one ball between the

**Table 1 Clinical data.**

| Patient | Sex | Age | SOZ according to iEEG | Surgery | Outcome at 1 year—Engel class |
|---|---|---|---|---|---|
| P1 | F | 16 | Diffuse. PF/P | VA of DNET lesion | I |
| P2 | M | 45 | Diffuse. B PF | NA | NA |
| P3 | F | 28 | L MTL[a] | NA | NA |
| P4 | F | 24 | R PF | Resection—R PF | II |
| P5 | M | 49 | L MTL[a] | VA—L MTL | II |
| P6 | F | 49 | L MTL[a] | RNS—L MTL | III |
| P7 | M | 38 | B MTL[a] | RNS—B MTL | III |
| P8 | F | 43 | R OF and L MTL[a] | RNS—R OR & L MTL | III |
| P9 | M | 69 | L T | VA—L T | IV |
| P10 | M | 17 | R rostral CC | Resection—R F tip/CC | I |
| P11 | M | 27 | L PF | Resection—L prefrontal | III |
| P12 | M | 35 | R T tip/OF | Resection—R T tip/OF | IV |
| P13 | M | 26 | Diffuse. PF/P | NA | NA |
| P14 | M | 27 | Diffuse. PF/P | NA | NA |

*R* right, *L* left, *B* bilateral, *PF* prefrontal, *OF* orbitofrontal, *P* parietal, *T* temporal, *CC* cingulate cortex, *MTL* mesial temporal lobe, *DNET* dysembryoplastic neuroepithelial tumor, *SOZ* seizure onset zone, *RNS* responsive neuro stimulation, *VA* visualase laser ablations.
[a]Contain neurons within the SOZ.

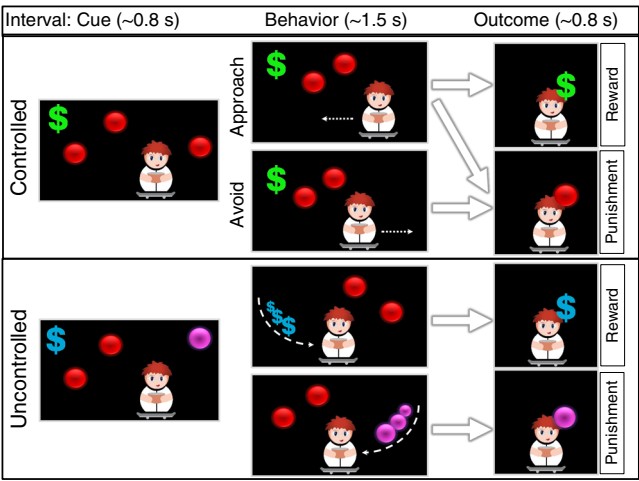

**Fig. 1 The paradigm.** The goal of the game was to earn virtual money by catching shekel signs and avoiding balls. A small avatar on a skateboard was located at the bottom of the screen and subjects had to move the avatar right and left using right and left arrow keys, in order to catch the money and avoid the balls falling from the top of the screen. There were two ways to gain or lose money—a "controlled" condition, where players actively approached green money signs (marked here as dollar signs) and avoided red balls, and an "uncontrolled" condition, where although cues appeared on the top of the screen (reward—cyan dollar sign, punishment—magenta ball), they always hit the avatar with no relation to the players' action (they chase the avatar during their fall). Each money catch resulted in a five-point gain and each ball hit resulted in a loss of five points, regardless of controllability (the outcome was shown on the screen after each trial). All four outcome event types occurred roughly at the same frequency, adaptive to the player's behavior. Each money trial was separated by a jittered interstimulus interval (ISI), which varied randomly between 550 and 2050 ms.

avatar and the rewarding cue). Here we focus on behavior during HGC, since previous results from this task showed differential activation in reward circuitry (ventral striatum) during this condition as well as an effect for individual differences[22].

Responses to motivational outcomes (Rewards or Punishments) from single and multi-units are recorded in the MTL, from the Amygdala ($N = 79$) and Hippocampus ($N = 61$) and in the mPFC, from the dorsomedial prefrontal cortex (dmPFC, $N = 63$) and cingulate cortex (CC, $N = 107$). Unit activity is analyzed with respect to outcome occurrence, evaluated for outcome valence specificity, controllability effect and the relation to subsequent behavioral choice (i.e., approach probability when facing a reward under HGC). We find that when players have control over the outcome, units in mPFC and MTL areas demonstrate a complementary role in the encoding of punishment and the affect on subsequent behavioral choice toward a reward cue. Specifically, while mPFC neurons selectively encode the negativity of motivational outcomes, relating neural responses to subsequent behavioral choices under high-conflict seemed to be the responsibility of neurons in the MTL (hippocampus and amygdala). Intriguingly, this cross-region outcome selective effect does not appear when participants had no control over the motivational outcome.

## Results

**Behavioral.** Similar to our previous findings in healthy populations[22], subjects showed an overall tendency to approach the rewarding cues on most trials (89.4% of 3285 trials) but less so

when they were presented under HGC (83.4 ± 10.4% approach, mean $N_{trials} = 104$ per patient) compared with LGC trials (94.6 ± 2.7% approach, mean $N_{trials} = 115$ per patient, Supplementary Fig. 1) [$t(14) = 4.78$, $p = 0.0003$, mean difference = 0.11, CI = (0.06, 0.16), Cohen's $d = 4.9$]. Mean response times were found to be significantly lower for HGC trials (807.05 ± 151.2 ms) compared with LGC trials (902.08 ± 160.3 ms) [$t(14) = 3.52$, $p = 0.003$, mean difference = 95, CI = (37.2, 152.9), Cohen's $d = 0.92$] (see Supplementary Fig. 1). Shorter reaction time during the HGC condition may be a result of task-related demands, as a faster response is necessary to avoid punishment when facing multiple threats.

Approach tendencies did not differ between patients with an MTL seizure onset zone (SOZ) (five patients, 84.7% and 91.1% approach for HGC and LGC, respectively) and patients with an outside MTL SOZ (nine patients, 80.8% and 92.7% approach for HGC and LGC, respectively) [Mann–Whitney test, $U = 19.5$, $Z = 0.61$, $p > 0.05$ for HGC and $U = 19$, $Z = 0.07$, $p > 0.05$ for LGC].

A generalized linear mixed model (GLMM) with subsequent HGC behavior as the dependent variable found no effect of: behavior on the current trial, movement, outcome (achieved or missed coin), and number of balls hitting the avatar on the way to the coin. Similarly, we did not find a significant behavioral- or paradigm-related effect of Punishment outcomes on behavior in subsequent HGC trials. Furthermore, no effect was found for the time lag between Punishment and subsequent HGC trials or movement in a period of 1 s before or after Punishment outcomes.

**Neuronal response selectivity to outcomes.** Neurons were considered responsive to a specific outcome condition (i.e., Controlled Reward, Uncontrolled Reward, Controlled Punishment, and Uncontrolled Punishment) if they significantly changed their firing rate (FR) following that outcome (between 200 and 800 ms post outcome occurrence, evaluated using a bootstrapping approach, see "Methods"). We found 31 of 79 (39%), 26 of 61 (43%), 26 of 63 (41%), and 46 of 107 (43%) neurons that significantly responded to at least one of the four outcome conditions in the MTL; Amygdala, Hippocampus, and mPFC: dmPFC, CC, respectively (see Fig. 2 for examples of neuronal selective FR).

To assess the sensitivity of neurons to the ability of players to control the outcomes, we examined their response probability to Controlled and Uncontrolled outcomes across valence type (Rewards and Punishments). A higher probability of responding to the Controlled outcomes over Uncontrolled outcomes was apparent in neurons from all four areas (Fig. 3a) [McNemar's exact test: MTL; Amygdala $\chi^2 = 9.3$, $p = 0.0088$; Hippocampus $\chi^2 = 3.3$, $p = 0.07$ ($\chi^2 = 13.8$, $p = 0.0004$ for MTL combined), mPFC; dmPFC $\chi^2 = 7.7$, $p = 0.011$; CC $\chi^2 = 4.3$, $p = 0.049$ ($\chi^2 = 12.25$, $p = 0.0005$ for mPFC combined), $p$ values were corrected for false-discovery rate (FDR)]. No main effect was found for valence in any of the recording areas (Fig. 3b). During the Controlled condition, mPFC neurons appeared more responsive to Controlled Punishments over Controlled Rewards (17 vs. 4 in the dmPFC and 20 vs. 10 in the CC), while in the MTL the response probability was similar for both types of valence (Fig. 3c; 12 vs. 12 in the Amygdala and 8 vs. 10 in the Hippocampus) [$\chi^2 = 7.2$, $p = 0.065$ for the four areas and $\chi^2 = 6.04$, $p = 0.014$ comparing MTL to mPFC]. This was not observed during the Uncontrolled condition (Fig. 3d). No such selectivity was observed for neurons in both MTL regions, even when removing neurons within the MTL SOZ (see Supplementary Note 2).

To examine the magnitude of neuronal selective responses we further calculated normalized FR changes separately for neurons with a significant increase in FR and neurons with a significant decrease in FR in at least one of the four outcome conditions,

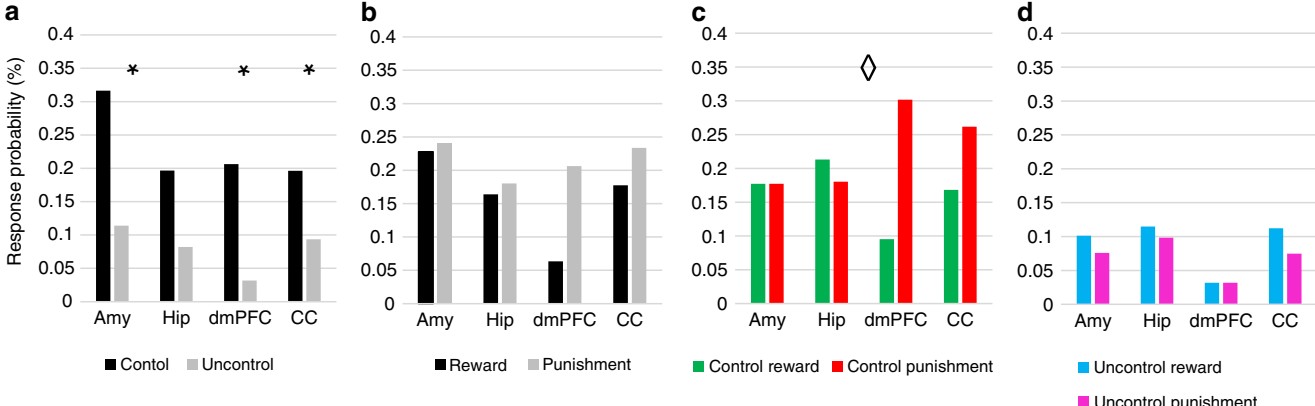

**Fig. 2 Example neural responses to the different outcome conditions. a** Sagittal slices show the location of active electrode contacts in mPFC and MTL areas (yellow markers, registered to an MNI atlas) that refer to the raster plots and peri-stimulus time histograms (PSTH), per condition (see legend for color codes). **b**, **c** PSTH from two different amygdala neurons showing significant increase in firing following reward outcome in both controlled and uncontrolled conditions. **d**–**f** PSTH from three different mPFC neurons showing significant increase in firing following punishment outcome in controlled but not in uncontrolled condition. Time 0 on the x-axis represents the timing of outcome (coin or ball hit the avatar). FR firing rate.

**Fig. 3 Neurons' response probability in different regions and outcome conditions.** Percent of neurons per region presenting a significant change in firing rate (FR) between 200 and 800 ms in response to: **a** Controlled (black) or Uncontrolled (gray) outcomes (across valence); N = 79, 61, 63, and 107 independently sampled neurons for the amygdala, hippocampus, dmPFC, and CC, respectively. A two-sided McNemar's exact test found effects at p = 0.088, p = 0.011, and p = 0.049 for the amygdala, dmPFC, and CC, respectively, FDR corrected. Asterisks represent significant at p < 0.05. **b** Reward (black) or Punishment (gray) outcomes (across controllability); N is similar to (**a**). **c** Controlled rewards or punishments (N = 93 independently sampled neurons from four or two brain regions, p = 0.065 and p = 0.014 using $\chi^2$ test, respectively). Diamond denotes significant valence preference between MTL and mPFC at p < 0.05. **d** Uncontrolled rewards or punishments (N = 39 independently sampled neurons from four or two brain regions, p = 0.97 and p = 0.99 using $\chi^2$ test, respectively). Amy amygdala, Hip hippocampus, dmPFC dorsomedial prefrontal cortex, CC cingulate cortex.

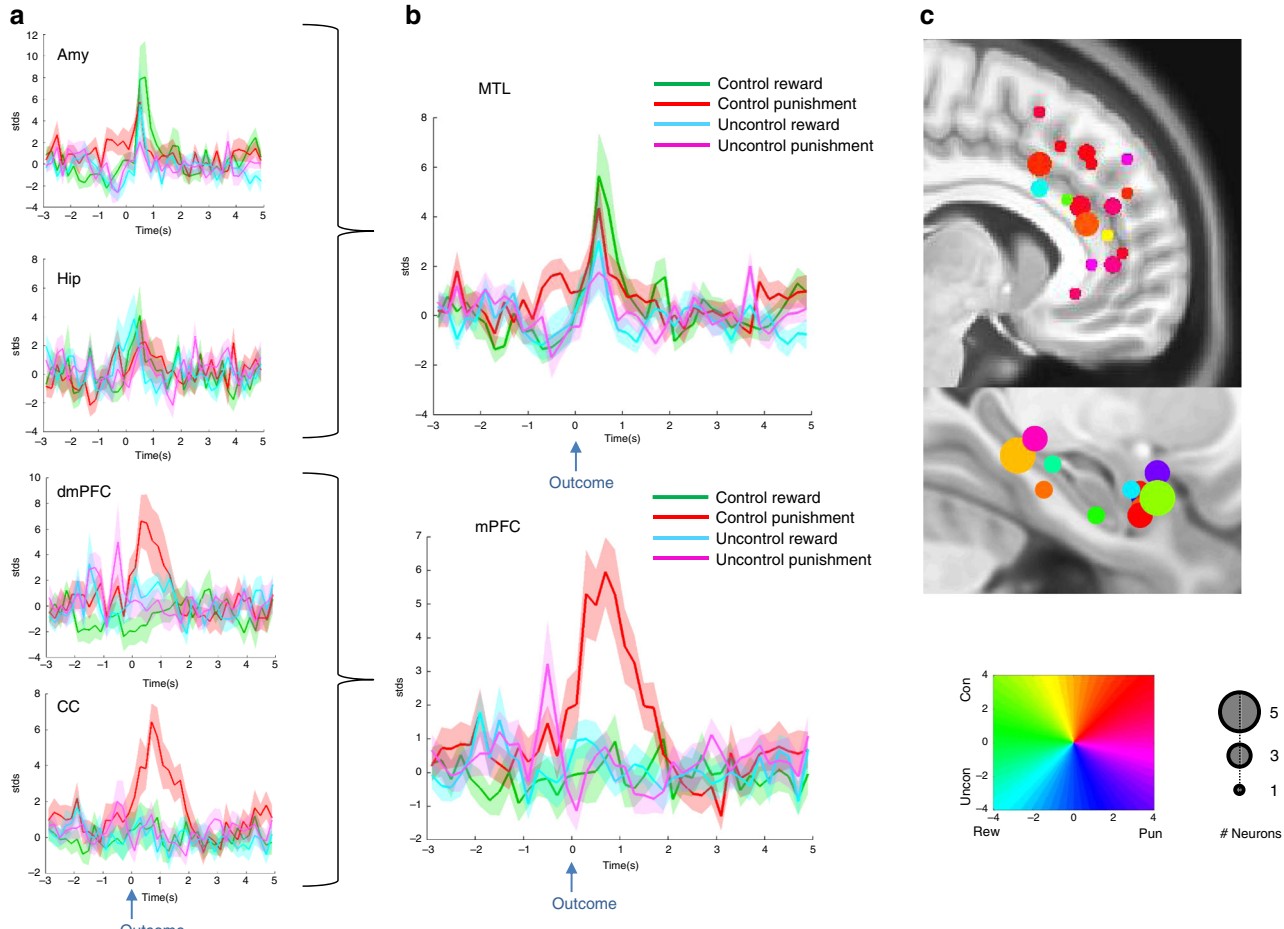

**Fig. 4 Selectivity of neural time-course responses per outcome types.** Average normalized FR for positively responsive neurons shown for (**a**) each of the four recording regions $N = 12, 9, 11$, and 22 for Amygdala, Hippocampus, dmPFC, and CC, respectively. **b** Combined regions in MTL and mPFC groups, $N = 24$ MTL and 34 mPFC neurons. Shaded area corresponds to standard error of mean (SEM). Source data are provided as a Source Data file. **c** Responsivity profile projected on two sagittal atlas slices for mPFC (upper panel) and MTL (lower panel) region groups. Coloring is according to the averaged normalized FR change for each condition (coloring key is presented in lower square). Circle size corresponds to the number of neurons from each contact groups, from 1 (smallest) to 5 (largest) (see key). Time 0 on the x-axis represents the timing of outcome (coin or ball hit the avatar). Amy amygdala, Hip hippocampus, dmPFC dorsomedial prefrontal cortex, CC cingulate cortex, MTL medial temporal lobe, mPFC medial prefrontal cortex, STDs standard deviations, Rew reward, Pun punishment, Uncon uncontrol, Con control.

excluding neurons with a mixed response (13% of responsive neurons: 6 in MTL and 11 in mPFC). Figure 4 presents the results obtained from this analysis for neurons with a significant increase in FR in response to outcomes. Overall, in line with the probability of FR, this analysis shows that there was greater selectivity in average response to outcome valence under Controlled conditions, more so in mPFC regions than in MTL regions (Fig. 4a). In light of the similarity in response selectivity for electrodes situated in areas within mPFC and within MTL, in further analyses we combined neurons from the amygdala and hippocampus to form an MTL neural group (140 neurons) and neurons from the dmPFC and CC to form the mPFC neural group (170 neurons).

A repeated measures ANOVA with normalized FR increase following outcome (200–800 msec) as the dependent variable and region groups [MTL, mPFC], controllability (Controlled/Uncontrolled) and outcome valence (Reward/Punishment) as the independent factors, revealed a greater response to Controlled Punishment outcomes, specifically in the mPFC region group (three-way interaction [$F(1, 56) = 13.6$, $p = 0.001$, $\eta^2 = 0.2$] demonstrated in time-course graphs in Fig. 4b). The ANOVA

further showed that the negative bias in response to outcome was more pronounced in mPFC neurons (two-way interaction of valence and region [$F(1, 56) = 5.6$, $p = 0.021$, $\eta^2 = 0.09$]), and that the preferred response to Controlled outcomes was more pronounced for negative valence (two-way interaction of valence and control [$F(1, 56) = 7.72$, $p = 0.007$, $\eta^2 = 0.12$]). A main effect for controllability showed higher FR in response to Controlled [mean = 3, CI = (2.2, 3.8)] compared with Uncontrolled outcomes [mean = 1.1, CI = (0.6, 1.5)] in both region groups [$F(1, 56) = 22.44$, $p < 0.001$, $\eta^2 = 0.29$]. The results were still significant when removing neurons within the SOZ from the analysis. Analyzing each region group separately, we found that MTL neurons displayed higher FR to Controlled [mean = 3.2, CI = (1.9, 4.6)] over Uncontrolled [mean = 1.6, CI = (0.7, 2.4)] outcomes [$F(1, 23) = 6.3$, $p = 0.02$, $\eta^2 = 0.22$]. In mPFC neurons, we found a significantly higher FR in response to Controlled [mean = 2.8, CI = (1.7, 3.8)] vs. Uncontrolled [mean = 0.6, CI = (0.2, 1.1)] outcomes [$F(1, 33) = 19.2$, $p = 0.001$, $\eta^2 = 0.37$], Punishment [mean = 2.9, CI = (1.9, 3.9)] vs. Reward [mean = 0.4, CI = (−0.7, 1.5)], [$F(1, 33) = 9.37$, $p = 0.004$, $\eta^2 = 0.22$], as well as a significant interaction [$F(1, 33) = 18.8$, $p < 0.001$, $\eta^2 = 0.36$]

resulting from a higher response to Controlled Punishment over the other three conditions ($p < 0.001$). Altogether these results suggest that although all neurons showed greater responsivity to Controlled outcomes, mPFC neurons exhibited significant selectivity to negative outcomes when players had control over the trial (shown graphically per region group and recording site in Fig. 4c). Supplementary Fig. 2 presents the results for FR decreases. A repeated measures ANOVA (with similar variables and factors as above) revealed a main effect of controllability [$F(1, 51) = 11.6$, $p = 0.001$, $\eta^2 = 0.19$], where Controlled trials evoked stronger decreases in FR [mean $= -2.2$, CI $= (-2.8, -1.6)$] compared with Uncontrolled trials [mean $= -0.8$, CI $= (-1.4, -0.2)$] in both region groups. No other main effect or interaction was significant.

To account for players' movements during the game, we performed a separate analysis while balancing trials across conditions according to the amount of key presses during each trial. The mPFC sensitivity to Punishment and Controllability did not seem to result from motion planning or artifacts, as evident by the similar results (Supplementary Fig. 4). mPFC neurons responded more to Control Punishment outcomes and MTL neurons more to Control Reward ($\chi^2 = 4.23$, $p = 0.04$). An increased FR was observed following Controlled Punishment vs. Uncontrolled Punishment in the mPFC, and was maintained following movement balancing [sign test, $Z = 2.6$, $p = 0.009$, FDR corrected]. An increased FR was also observed following Controlled Punishment vs. Controlled Reward in the mPFC [sign test, $Z = 3.36$, $p = 0.003$, FDR corrected]. In contrast, a higher MTL FR during the Controlled as compared with the Uncontrolled condition was not significant after controlling for movements.

mPFC selectivity to Controlled Punishment over Controlled Reward and Uncontrolled Punishment seems to be a general phenomenon regardless of whether punishments were obtained when a reward was chased (an unsuccessful approach trial) or when there was no reward present at all (Supplementary Fig. 5). For example, 21 mPFC neurons exclusively responded to Punishments without Rewards on the screen compared with 8 neurons exclusively responsive to Rewards, and 14 mPFC neurons exclusively responded to Punishments during an unsuccessful approach trial compared with 8 neurons exclusively responsive to Rewards (the same result was found when comparing to Uncontrolled Punishment, see Supplementary Table 3). In contrast, there was no such difference observed in MTL neurons (it should be noted that Punishments can also be obtained during failed avoidance trials, but such events were rare and could not be evaluated).

Lastly, we examined the relative timing of response to outcome in each region group per outcome type. Overlapping the time courses revealed earlier responses in mPFC compared with MTL neurons following outcome (Fig. 5a). These were only evident in the Controlled conditions, where responses were already significantly above baseline at 0–200 ms following Punishment for mPFC neurons [signed rank, $Z = 346$, $p < 0.05$, FDR corrected] and only 200–400 ms for MTL neurons [signed rank $Z = 78$, $p < 0.05$, FDR corrected]. Responses to Reward were significantly above baseline already at 0–200 ms for mPFC neurons [signed rank, $Z = 43$, $p < 0.01$, FDR corrected] and only 400–600 ms for MTL neurons [signed rank $Z = 102$, $p < 0.01$, FDR corrected]. In the Uncontrolled trials (Fig. 5a) responses were overall not significantly above baseline at any of the 200 ms epochs we measured following outcome with the exception of the 200–400 ms window in mPFC neurons during the Uncontrolled Reward condition [signed rank $Z = 36$, $p = 0.02$, FDR corrected].

**Neuronal response to outcome effects on subsequent behavioral choice.** To test for brain-behavior interactions we assessed behavioral approach tendency with respect to neuronal firing in the previous trial. When tested with respect to HGC Controlled trials we found a distinct effect in the MTL neurons. We concentrated on evaluating the effect on behavior during HGC trials because approach probability during LGC trials was very high (92%). Neuronal firing following Controlled Punishment outcomes correlated with decreased probability for approach behavior in the next Controlled trials, whereas firing following Controlled Reward outcomes correlated with increased probability for approach behavior in subsequent Controlled trials [Mann–Whitney test, $U = 13$, $Z = -2.9$, $p = 0.01$, FDR corrected]. Neurons in the mPFC did not present such a differential effect (Fig. 5b, c). Neural responses to both types of Uncontrolled outcomes in the MTL and mPFC were not predictive of subsequent approach behavior in the following Controlled HGC trials (see Supplementary Fig. 3).

To further evaluate the complex interaction between the observed phenomenon and other paradigm-related variables we performed six GLMM (binomial), with behavior in subsequent HGC trials as the dependent variable and evaluating each of: Punishment temporal, Punishment frontal, Reward temporal, Reward frontal, Punishment interaction, and Reward interaction as independent variables. We found that only MTL firing following Punishment outcomes significantly correlated with behavior in subsequent HGC reward trials [beta $= 1.1$, $t = 4.22$, $p < 0.0001$, FDR corrected], even after accounting for movement and time between Punishment and subsequent HGC trials. Even when removing MTL neurons (two responsive neurons from the left amygdala of patient 6) that were within the SOZ, this finding remained significant [beta $= 1.2$, $t = 4.3$, $p < 0.0001$, FDR corrected]. This result was not replicated for the LGC trials; MTL response to punishment did not predict subsequent behavior under LGC. Breaking this result down into the different structures, we found that this was significant in the Hippocampus [beta $= 1.25$, $t = 3.2$, $p = 0.006$, FDR corrected] but not in the other regions: Amygdala, dmPFC, and CC.

A similar analysis on neural firing following Reward compared subsequent HGC behavior with the previous HGC trial, after accounting for the previous HGC-related variables: movement, behavior outcome (achieved or missed the coin), and number of ball hits. However, we did not find that mPFC or MTL response correlated with behavior in subsequent HGC trials. A similar analysis for LGC also failed to show an effect of neural response to outcome on subsequent trials.

To evaluate the association between mPFC responsivity to Control Punishment on the one hand and the subsequent behavioral effect of MTL neurons to Punishment on the other hand, we focused on four sessions that had increased firing neurons in both region groups. We found that the interaction between regions' firing following Controlled Punishment was predictive of subsequent HGC behavior [beta $= 12.19$, $t = 3.14$, $p = 0.0018$, FDR corrected].

## Discussion

The present study applied intracranial recordings from neurons in the mPFC (dmPFC and CC) and MTL (amygdala and hippocampus) of humans while they participated in an ecological goal-conflict game situation. We now present evidence of timed involvement of MTL and mPFC neurons in integrating outcome valence and its effect on subsequent goal-conflict resolution. Our results show that neurons in the mPFC areas were more sensitive to Punishment than Reward outcomes, but only in the game

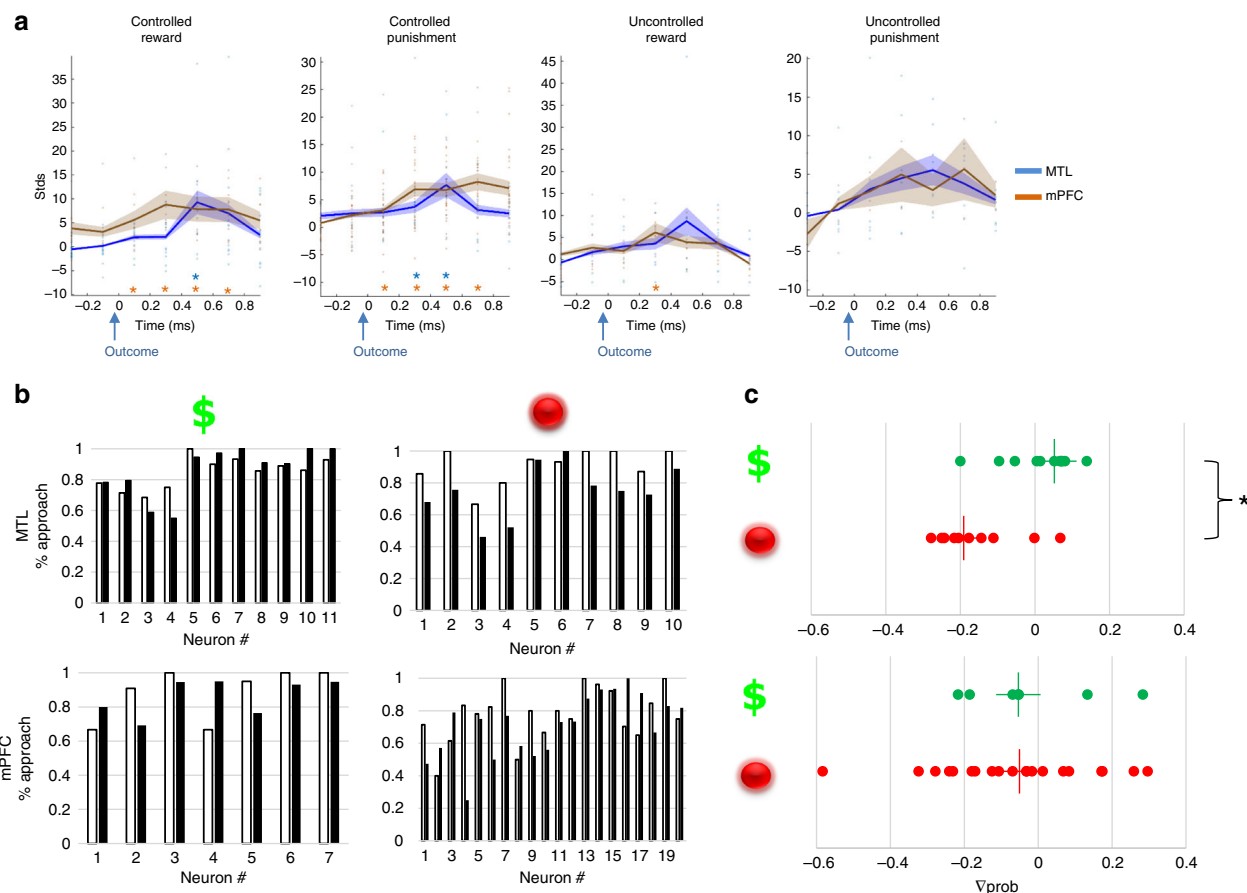

**Fig. 5 Neural firing change and their effect on subsequent behavioral choice. a** Time courses for mean normalized FR change in mPFC (brown trace) and MTL (blue trace) sites for each outcome type. Asterisks mark for which 200 ms window the FR is significantly above baseline (two-sided signed rank test, $p < 0.05$ FDR corrected, Control Reward $N = 14$ and 9 for MTL and mPFC, respectively). Control Punishment $N = 12$ and 29 for MTL and mPFC, respectively. Uncontrol Reward $N = 8$ and 8 for MTL and mPFC, respectively. Uncontrol Punishment $N = 8$ and 2 for MTL and mPFC, respectively. Note that for both controlled outcomes mPFC neurons fire slightly before MTL neurons (left) but not for uncontrolled outcomes (right). Time 0 on the x-axis represents the timing of outcome (coin or ball hit the avatar). Shaded area corresponds to SEM. Source data are provided as a Source Data file. **b, c** The effect of neural responses following controlled outcomes on subsequent behavioral choice under HGC condition. **b** The mean probability for approaching a coin, subsequent to trials where a neuron fired 200–800 ms following a controlled outcome (black bars) vs. trials where a neuron did not fire (white bars), shown for MTL (**b**, top) and mPFC (**b**, bottom) neuron groups per outcome type. Note that only MTL neuron showed a consistent pattern of subsequent behavior of less approach after punishment outcome (**b**, top right). **c** Approach probability change following neural firing to controlled rewards (green markers) and controlled punishments (red markers) in MTL (**c**, top) and mPFC (**c**, bottom). **c** (top) Asterisk denotes a significant two-sided Mann–Whitney test, $p = 0.01$, FDR corrected, $N = 21$ neurons (10 punishment, 11 reward). STDs standard deviations, Delta Prob difference in probability.

periods in which participants had a choice and could control the outcome with their behavior (i.e., Controlled condition). Compared with mPFC, the MTL showed smaller preference to Controlled outcomes, but without bias to outcome valence (Figs. 3 and 4). Yet, despite this apparent valence blindness, MTL firing following Controlled Punishment outcomes was associated with decreased approach probability when faced with a high-conflict situation in the next trial (HGC trials, Fig. 5). Although mPFC neurons alone did not show such a direct association with behavioral choice, the interaction of mPFC with MTL neuronal responses to Controlled Punishment outcomes decreased subsequent approach probability.

The bias of neurons in the dmPFC and CC to encode negative outcome is in line with previous studies showing the involvement of these regions in processing pain[23] and economic loss[24]. As these regions are also involved in motion planning and inhibition[25]; such negative bias in the context of goal-directed behavioral choice could be explained by the critical need to reduce false negatives for survival[26]. This evolutionary rationale is

supported by our finding that negative neuronal bias was only apparent when players had control over the outcome. This, however, stands in contrast to results reported by Hill et al.[27] showing neural sensitivity to positive over negative outcomes (wins vs. losses) in the human mPFC. In this study, participants had to choose between two decks of cards with positive or negative reward probabilities and values. Thus, in contrast to our task, they were not directly faced with conflicting goals but rather had to learn the probability of positive and negative outcomes. Moreover, they did not have to actively move toward or away from a goal. These differences represent a major contrast between reinforcement learning tasks and anxiety-related approach-avoidance paradigms that are more similar to our task. the immersive nature and call for action of approach-avoidance scenarios in our study may bias sensitivity to punishing threats over desired rewards. One could argue that the negative outcome sensitivity shown by the mPFC neurons could be accounted for by an inhibition of prior approach behavior rather than to the response to punishment itself. To refute this possibility we show

that the valence selectivity of FR in the mPFC is unrelated to the different types of punishments; occurring either with or without the presence of a reward (see Supplementary Fig. 5).

This finding that MTL were responsive to both reward and punishment outcomes is consistent with known involvement of amygdala and hippocampus in both positive and negative emotion processing in a motivation-related context[6,28,29]. For example, Paton et al.[6] found that distinct amygdala neurons respond preferentially to positive or negative value.

A central finding in this study is the role of MTL neurons in reduced approach choice following negative outcome. Further examination of this finding in the various MTL structures showed that this result primarily stems from the hippocampal neurons, as their effect on subsequent behavioral choices was significant. The Reinforcement Sensitivity Theory (RST) of Gray and McNaughton[9] proposed that the hippocampus, as part of the behavioral inhibition system, is in charge of resolving goal-conflict situations mediating the selection of more adaptive behaviors according to the acquired motivational significance. More recently, fMRI studies in humans supported such a role of the hippocampus in goal-directed gambling tasks[30,31]. For example, Gonen et al.[31] applied dynamic causal modeling to fMRI data showing that the hippocampus received inputs regarding both positive and negative reinforcements, while participants decided to take a risk or play safely in a computerized gambling game.

However, diverging from the RST model, our results point to the significance of the MTL, not only in the online processing of positive and negative reinforcements but also in the use of such information to influence future motivation behavior. This fits well with the hippocampus' known role in association learning and extinction[32]. Unfortunately, our design did not allow an objective evaluation of neural response directly following cue appearance due to unbalanced trials across the different conditions (see Supplementary Note 1), resulting in confounding saliency effects. Future studies with a similar design but balanced cue trials are warranted to evaluate the MTL's role during the decision-making phase.

In addition, classical reinforcement learning studies have highlighted the important role of PEs, in the striatum, in providing a learning signal to update subsequent behavior. Recent studies, however, have shown that the hippocampus and the striatum interact cooperatively to support both episodic encoding and reinforcement learning[33–35]. Thus, it is interesting to observe that converging evidence from different study cohorts, including goal conflict, reinforcement learning and memory, seem to point to the important role of the MTL, and hippocampus in particular, in learning from outcomes to update behavior.

In a subsample of our data, the interaction of mPFC and MTL neuronal activity following punishment was significantly predictive of subsequent avoidance. This finding corresponds with a line of recent animal studies showing that inputs from the hippocampus and/or amygdala to the mPFC underlie anxiety state and avoidance behavior[36,37]. For example, theta (4–12 Hz) synchrony emerged between the vHPC and mPFC during rodents' exposure to anxiogenic environments[38]. Moreover, single units in the mPFC that synchronized with the vHPC theta bursts, preferentially represented arm type in the EPM[15]. Further analysis in humans could test for the relation between hippocampus-mPFC theta synchrony and unit activity in the hippocampus.

The combined evidence from animal studies and our findings in humans, suggest that the MTL, and the hippocampus in particular, play an important role in updating approach tendencies after receiving a signal of negative outcome value from the mPFC. This temporal sequence though seems to contradict the anxiety-related animal models described previously. However, these studies often apply the EPM and related paradigms that cannot dissociate in time the acquisition and updating of approach tendencies following the outcome phase from the behavioral decision-making phase.

It has been widely acknowledged that the hippocampus, amygdala, and mPFC share anatomical and functional connectivity as a distributed network that supports anxiety behavior in an interdependent manner, and that mPFC to MTL innervation exists and is related to approach-avoidance tendencies[15,37]. The evidence also shows that the leading direction of such connections is context dependent[39,40]. We speculate that in the outcome evaluation phase and before the next behavioral choice, the MTL is responsible for storing their motivational significance for future decisions under goal conflict, using inputs received from a number of cortical and subcortical nodes, including negative value signals from the mPFC following punishments. Conversely, during the actual goal-conflict behavior, an inhibiting approach could be a more direct product of mPFC activity, dependent on MTL updated inputs[36,41]. Neural dynamics during the decision phase in our paradigm was difficult to assess due to excessive movements and rapidly changing contexts (balls falling continuously) and further studies are warranted.

Remarkably, both the valence selectivity of mPFC neural responses and the effect of MTL outcome responses on subsequent behavior were evident only during Controlled condition (see Fig. 2). It has been argued that the neural response to outcome value and valence, as well as subsequent goal-directed behavior, is influenced by one's sense of control over a given situation—often referred to as the process of agency estimation[42–44]. Thus, one's sense of control may play a role in future motivational behavior[45]. Specifically, it has been suggested that a sense of control can bias the organism toward a proactive response, encouraging it to optimize approach-related decisions while giving more weight to certain outcomes. For example, a diminished sense of control as seen in depression may prevent one from learning adaptive behavior toward rewards, despite an intact ability to assign motivational significance to the goals[46]. Another example is PTSD, where an exaggerated sense of agency over a traumatic event is suggested to intensify the negative value attached to even distant reminders of the traumatic event, resulting in maladaptive avoidance behavior, even when motivational significance is realized[46,47].

Intriguingly, recent imaging studies show that agency estimation relies on activations of the mPFC, particularly its dorsal aspect (e.g., the supplementary motor area (SMA), pre-SMA, and dorsomedial PFC)[48]. Further studies are needed to evaluate whether the findings observed here relate to agency estimation processing in the mPFC or reflect the effect of another region (such as the angular gyrus[49]) on motivational processing in mPFC.

Our data were obtained from patients with epilepsy and therefore the generalization of these results to other populations should be considered with caution. It has been previously suggested that patients with temporal lobe lesions present more approach behavior compared with controls[30]. In addition, studies using Stroop-related paradigms showed that patients with MTL lesions present impaired performance on conflict tasks[50,51]. Other studies, however, found no major difference in the Stroop task between MTL patients and healthy controls[52]. We believe our findings are not specific to MTL lesions or epilepsy for several reasons. First, only 5 of the 14 patients had seizures originating from the MTL, and these five patients did not exhibit different approach tendencies compared with the extra-MTL patient group. Second, removing the few neurons from within the epileptic SOZ in the MTL did not change the significance of the results, either neuronally or behaviorally. Lastly, approach probabilities and reaction times obtained from our group of patients

were similar to those obtained from a control group of 20 healthy participants (see Supplementary Note 2). To evaluate this further, future studies should adopt similar ecological procedures using noninvasive imaging methods (e.g., EEG, NIRS, or fMRI).

Unfortunately, the ecological nature of our paradigm does not allow the evaluation of neural responses at timings prior to outcome (i.e., anticipation), since this time period is contaminated by movements and simultaneous occurrence of different events (rewards and punishment).

In addition, neurons from different substructures, such as the ventral and dorsal hippocampi, basolateral and central amygdala, were aggregated. It is known that these substructures play a different and sometimes contradictory role in motivational processes[53].

Lastly, in an attempt to increase participants' engagement, players were encouraged to obtain the rewards, which resulted in a high approach probability. Due to the sparseness of avoidance behavioral choices, avoidance trials were not analyzed by themselves for neural responsivity. It would be of interest to evaluate neural response to avoidance in future studies. However, despite approach dominance, in a previous fMRI study with this paradigm[22], we found marked differences in behavior and brain responses between HGC and LGC conditions, which lead us to conclude that there is significant goal conflict under the HGC condition. First, there was less approach behavior under HGC than LGC trials. Furthermore, brain mapping analysis during approach under HGC vs. LGC conditions showed greater mesolimbic BOLD activity and functional connectivity under HGC. Lastly, individual differences in approach/avoidance personality tendencies (indicated by standard personality questionnaires) revealed that individuals with approach personality tendency showed more approach behavior during HGC trials than individuals with avoidance-oriented personality. Altogether, these findings support our operationalization of the HGC condition.

In summary, our findings suggest differential process specificity for the MTL and mPFC following goal-directed behavior under conflict. The mPFC showed response sensitivity to the integration of negative outcomes under a controlled condition, possibly reflecting the importance of a sense of agency on assigning value to outcomes. In contrast to mPFC, the MTL neurons showed minimal response selectivity to valence of outcome, yet following punishment their responses modified approach behavioral choices under high conflicts. These findings point to the important role of MTL, and the hippocampus in particular, in learning from outcomes in order to update our behavior, a major issue in mental disorders such as addiction and borderline personality disorders. Future studies should evaluate how this differential processing could assist in computational modeling of psychiatric disorders as well as assigning process-specific targets for brain-guided interventions.

## Methods

**Participants**. Fourteen patients with pharmacologically intractable epilepsy (nine males, 35.2 ± 14.6 years old) participated in this study. Ten patients were recorded at the Tel Aviv Sourasky Medical Center (TASMC) and four at the University of California Los Angeles (UCLA) with similar experimental protocols and recording systems. One patient (patient 4) underwent two separate implantations with a time lag of 6 months. A total of 20 sessions were recorded. Patients were implanted with chronic depth electrodes for 1–3 weeks to determine the seizure focus for possible surgical resection. The number and specific sites of electrode implantation were determined exclusively on clinical grounds. Patients volunteered for the study and gave written informed consent. The study conformed to the guidelines of the Medical Institutional Review Boards of TASMC and UCLA. An additional group of 20 healthy participants performed the task in a laboratory room (Supplementary Note 2). These participants volunteered for the study and gave written informed consent. The healthy participant study was approved by the Tel Aviv University Ethics Committee.

**Electrophysiology**. Through the lumen of the clinical electrodes, nine Pt/Ir microwires were inserted into the tissue, eight active recording channels and one reference. The differential signal from the microwires was amplified and sampled at 30 kHz using a 128-channel BlackRock recording system (Blackrock Microsystems) and recorded using the Neuroport Central software up to version 6.05. The extracellular signals were band-pass filtered (300 Hz to 3 kHz) and later analyzed offline. Spikes were detected and sorted using the wave_clus toolbox (version 1.1)[54] and MATLAB (mathworks, version 2018a). Units were classified by one of the authors (T.G.) based on spike shape, variance, and the presence of a refractory period for the single units[55]. Units were classified as putative single units and multi-unit clusters based on the presence of a refractory period (single unit had to present at least 99% of action potentials which were separated by an inter-spike interval of 3 ms or more) and based on spike shape. To anatomically localize single-unit recording sites we registered computerized tomography images acquired postimplantation to high-resolution T1-weighted magnetic resonance imaging data acquired preimplantation using SPM12 (http://www.fil.ion.ucl.ac.uk/spm). Micro Electrode locations with units can be seen in Fig. 2e. In total, we recorded 79 Amygdala, 61 hippocampus, 63 dmPFC, and 107 cingulate units. Microwire locations are described in Supplementary Table 1.

**Experimental design and sessions**. Subjects sat in bed facing a laptop and were asked to perform the Punishment, Reward, and Incentive Motivation game (PRIMO game[22]; Java1.6, Oracle, Redwood-Shores, CA & Processing package, http://www.processing.org). The goal of the game was to earn money by catching coins and avoiding balls. The monetary reward was virtual, no real money was delivered at any time to the participants. A small avatar on a skateboard was located at the bottom of the screen and subjects had to move the avatar right and left using the right and left arrow keys, in order to catch the money and avoid the balls falling from the top of the screen. There were two ways to gain or lose money —a "Controlled" condition, where players actively approached coins and avoided balls (which fall in a straight to zig-zag fashion from the top of the screen) and an "Uncontrolled" condition, where although cues appeared on the top of the screen, they hit the avatar randomly without relation to the players' action (Fig. 1). During Uncontrolled Reward or Punishment trials, controlled balls can appear along with the uncontrolled cue, but they are relatively easy to avoid as there is no conflict (the subject has the ability to move the avatar throughout the game regardless of the trial's condition). Each coin catch resulted in a five-point gain and each ball hit resulted in a loss of five points, regardless of controllability. To create an ecological environment, the difficulty level of the game was modified every 10 s according to the local and global performance of the player. By dynamically adjusting the difficulty level (speed) and actively balancing the number of Uncontrolled events, the game was tailored to match each player's skills and all Outcome event types occurred roughly at the same frequency. Each Reward trial was separated by a jittered interstimulus interval, which varied randomly between 550 and 2050 ms. To construct HGC and LGC, the number of obstacles (i.e., balls) placed between the player and the falling coin changed in each trial. Trials with 0–1 balls between the player and the falling coin were defined as LGC trials, while trials with 2–6 balls between the player and the coin were defined as HGC. The game was played for three or four blocks (according to the patient's agreement) of 6 min each, starting with 1-min fixation point. Subjects received instructions prior to playing the first session. Subjects 2, 4, 6, 8, 10, and 12 played the game twice during their monitoring period at a lag of 13, 5, 4, 2, 4, and 2 days between sessions, respectively. The paradigm was identical to that used in Gonen et al.[22] with one exception: the flying figure used in the previous version to signal Uncontrolled trials was removed (to avoid neural responses to its appearance). In the current version, the colors of the Rewards and Punishment were changed to signal non-Controllability. Uncontrolled rewards were in cyan color (vs. green for controlled rewards) and Uncontrolled balls were orange in color (vs. red for controlled balls). This was explained to the subjects during training.

**Analysis of behavioral data**. Once a Controlled Reward cue appeared at the top of the screen, subjects had to decide whether to approach it (at the risk of a possible hit by a ball) or to avoid it (and thus minimize the risk of getting hit). Controlled Reward trials were classified to approach and avoidance trials according to the player's behavior in each game session, based on a machine learning classification model[22].

**Analysis of neural data**. Data were analyzed using MATLAB (version 2018a). Raster plots were binned to nonoverlapping windows of 200 ms length to create FR per window and summed across trials to create peri-stimulus time histograms (PSTH). PSTHs were initially calculated for a period of 8 s from 3 s before outcome stimulus to 5 s post stimulus (40 windows). For evaluating neuron responsiveness, we concentrated on the time period of 200–800 ms post outcome stimulus (similarly to Ison et al.[56]). As this study concentrates on neural response to outcome, time 0 relates to the moment when the ball or coin hits the avatar (time of outcome), throughout the manuscript.

**Criteria for a responsive unit**. To evaluate neural responsiveness to the different conditions, we adopted a bootstrapping approach. Since the PRIMO game is

interactive and ongoing there is no distinct baseline period prior to each trial. We thus created a distribution of FR where each instance in the distribution is calculated as a FR average of N windows randomly selected from the entire session period, where $N$ is the number of trials of the specific condition. Thus, the only difference between the measured FR and such an instance is that the actual measured FR was time-locked to the events of the specific condition. The distribution was built from 1000 such instances and the neuron's window was considered positively responsive to the condition if the probability of obtaining the measured FR or higher was <0.01, and negatively responsive if the probability of obtaining the measured FR or lower was <0.01. A neuron was considered responsive to the condition if it was responsive in at least one of the three windows between 200 and 800 ms post event. Neural yield per area is described in Supplementary Table 2.

**Comparison between conditions**. To evaluate the main effect of controllability (over emotional value) we united reward and punishment trials and evaluated which neurons significantly changed FR following controlled or uncontrolled trials (with a similar bootstrapping approach). A similar analysis was performed for evaluating emotional value (over controllability). Next, we used chi-square analysis to assess whether the probability of responding to a specific condition varied between the different anatomical groups of neurons. In an additional analysis, we evaluated which neurons were positively responsive (increased FR) to at least one of the four conditions and which were negatively responsive to at least one condition. Neurons with a positive response to one condition and a negative response to another were excluded ($N = 4$, 2, 0, and 12 for Amygdala, Hippocampus, dmPFC, and CC, respectively). For both the positively responsive neurons and the negatively responsive neurons we computed a repeated measures ANOVA with region as the between subject variable and controllability and valence as the within subject variables and normalized FR of the different neurons as the dependent variable. Normalized FR was calculated by:

$$\bar{FR} = \text{mean}\left(\frac{FR_{window} - \bar{FR}_{random}}{\sigma_{random}}\right), \quad (1)$$

where $\bar{FR}_{random}$ is the average of $N$ 200 ms long randomly selected windows, $\sigma_{random}$ is the standard deviation of these randomly selected windows, and the mean is across the three windows (200–400, 400–600, and 600–800 ms post stimulus).

To evaluate whether the difference between frontal and temporal neurons is due to motions (either motion planning or artifact), we balanced motion (as obtained from the number of key presses) by excluding trials with high or low motion resulting in similar median motion scores (across remaining trials) between the two compared conditions. This analysis was performed separately for each condition pair.

Similarly, to evaluate the neural responses for different scenarios in which punishment was obtained, we balanced the number of trials in each paired comparison and tested the number of units that responded to each condition. The conditions for the paired comparisons included punishment without a reward present, punishment following a failed approach response, an uncontrolled punishment. These were also compared with a controlled reward outcome (see comparison results in Supplementary Table 3).

**Time course of neural data**. PSTHs were calculated for a period of 8 s, from 3 s before the stimulus to 5 s post stimulus (40 windows of length 200 ms each). The time course for each condition and region was created by averaging normalized FR (per window) across condition-specific positively responsive neurons during this time period. For each of the four windows between 0 and 800 ms, we evaluated whether the response to the specific outcome condition was significantly above baseline.

**Outcome—behavioral link**. To evaluate whether neural response to outcome affects future behavior, we analyzed each of the four outcome conditions separately. For each condition, we focused on neurons with a significant increase in FR following its outcome. For these neurons we divided outcome trials into trials with a neural response (neural firing between 200 and 800 ms following outcome) and trials without a neural response. Neurons with a high FR which resulted in less than one trial in which the neuron did not fire and followed by an approach choice or less than one trial in which the neuron did not fire and followed by an avoidance choice were omitted from this analysis. Next, for each trial, we evaluated whether the subsequent coin trial resulted in approach or avoidance behavior. Thus we could compare, for each condition, the effect of neural response following outcome on subsequent behaviors. Only outcome trials with a subsequent HGC coin trial (with more than one ball on the way to the coin) were included as LGC trials almost always resulted in approach behavior (above 94%).

To evaluate the complex interaction of neural firing, behavior- and paradigm-related variables, we performed six GLMM (binomial) with behavior in subsequent HGC trials as the dependent variable. GLMM test 1–2: for each HGC trial, we evaluated the previous punishment outcome by calculating the following variables: (1) a binary index indicating whether a temporal/frontal neuron fired in the time range 200–800 ms following punishment outcome (as before, only neurons that significantly increased FR following punishment outcome were evaluated); (2)

normalized total movement ±1 s of outcome time; (3) time delay between the punishment outcome and subsequent HGC trial. The analysis was done separately for temporal (GLMM test 1: 12 neurons, 953 total trials) and frontal neurons (GLMM test 2: 29 neurons, 2187 total trials) with neuron as the grouping variable. We used these three variables for both fixed and random effects including fixed and random intercepts grouping trials by neurons.

GLMM tests 3–4: for each HGC trial, we evaluated the previous HGC trial by calculating the following variables: (1) a binary index indicating whether a temporal/frontal neuron fired in the time range 200–800 ms following reward outcome (as before, only neurons that significantly increased FR following reward outcome were evaluated); (2) outcome—a binary variable indication whether the coin was caught or missed; (3) normalized total movement between reward appearance and disappearance either by avatar catching or missing; (4) number of ball hits on the way to the coin; and (5) behavioral decision (to approach or not). The analysis was done separately for temporal (GLMM test 3: 14 neurons, 1023 total trials) and frontal neurons (GLMM test 4: 19 neurons, 748 total trials) with neuron as the grouping variable.

To evaluate the connection between frontal responsivity to controlled punishment on one hand and the subsequent behavioral effect of temporal neurons to punishment on the other hand, we concentrated on four sessions (from patients 3, 4 and two sessions from patient 7) that had neurons with punishment-related FR increase in both the temporal and frontal lobes. For each HGC trial, we evaluated previous punishment calculating the following variables: (1) average firing of temporal neurons in the time range 200–800 ms following punishment outcome; (2) average firing of frontal neurons in the time range 200–800 ms following punishment outcome; (3) interaction between the previous variables; (4) normalized total movement ±1 s of outcome time; (5) time delay between the punishment outcome and subsequent HGC trial.

Similarly, for reward trials, we concentrated on three sessions (from patients 3, 7, and 10) that had neurons with reward-related FR increase in both the temporal and frontal lobes. For each HGC trial, we evaluated the previous HGC trial by calculating the following variables: (1) average firing of temporal neurons in the time range 200–800 ms following HGC reward outcome; (2) average firing of frontal neurons in the time range 200–800 ms following HGC reward outcome; (3) interaction between the previous variables; (4) normalized total movement between reward appearance and disappearance either by avatar catching or missing; (5) number of ball hits on the way to the coin (we did not add the behavior variable since all previous HGC trials in this case turned out to be approach trials).

**Statistics and reproducibility**. All experiments were only performed once. Source data are provided with this paper.

**Reporting summary**. Further information on research design is available in the Nature Research Reporting Summary linked to this article.

## Data availability
A reporting summary for this article is available as a Supplementary Information file. The source data underlying Figs. 4a, b, 5 and Supplementary Figs. 1, 2, 4, and 5 are provided as a Source Data file. Additional data are available from the corresponding author upon reasonable request.

## Code availability
Custom Matlab scripts are available through the following URL: https://github.com/tomergazit1/mPFC-and-MTL-neuronal-response-to-outcome-affects-subsequent-choice-paper-.

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

## Acknowledgements

We acknowledge financial support from the European Union Seventh Framework Program (FP7/2007-2013) under grant agreement no. 604102 (Human Brain Project). This work was also supported by the I-CORE Program of the Planning and Budgeting Committee and the Israel Science Foundation (grant no. 51/11, TH) and the Sagol Family Fund.

## Author contributions

T.H., T.Ga., and T.Go. conceived the study and designed the experiment. T.Ga., G.G., and N.C. analyzed the data. T.Ga., H.Y., and G.G. ran the experiments. I.F. and I.S. performed the surgeries and supervised the experiments and all aspects of data collection. Y.Z. assisted with statistical analyses. G.G. contributed to electrode localization. F.F. took care of the patients at TASMC. T.H. and I.F. supervised methodology and interpretation of findings. T.Ga., G.G., and T.H. wrote the paper. I.F., F.F., and T.Go. further contributed to the writing by reviewing and editing the manuscript.

## Competing interests

The authors declare no competing interests.
