## [Peer Review File · Nature Communications]

Reviewers' comments:

Reviewer #1 (Remarks to the Author):

This manuscript by Gazit and colleagues describes an interesting electrophysiological recording study of neural encoding of reward and punishment outcomes during a videogame in human patients who had received multiple electrode implantation for epilepsy.

The authors recorded in prefrontal cortex, amygdala and medial temporal lobe (hippocampus) sites as patients played a game in which they could either win or lose at particular moments. While the authors report that there were no main effects of outcome valence, suggesting relatively subdued coding of reward or loss outcomes overall, there were a number of valence/subregion interactions that are intriguing. In particular, prefrontal cortex had more far neurons that were exclusively responsive to punishment than to reward in a control condition. In other findings the authors report the amygdala and temporal lobe sites were relatively valence insensitive, though temporal lobe firing following punishment outcomes during controlled situations usually reflected increased avoidance behavior in high conflict situations. Also, there was increased firing rate during controlled conditions compared to uncontrolled conditions. A temporal pattern was also found in which prefrontal responses preceded temporal lobe findings.

A chief merit of this study is the relatively unique electrophysiological recordings afforded by the implanted patient group. This provides a rare window into firing patterns during the win/lose game.

The data patterns themselves are fairly subtle and complex, and do not easily give rise to strong interpretations. The authors do a good job in wrestling with interpretation, and their suggestion that prefrontal negativity bias could reflect an evolutionary importance of avoiding negative outcomes seems plausible. The authors also suggest that the temporal lobe, and hippocampus in particular, update avoidance tendencies after receiving a negative outcome signal from prefrontal cortex. All of this seems reasonable, and is consistent with much literature.

Overall, this is an interesting and unusual study. While the data are not particularly conducive to strong interpretation, the authors make a valiant attempt and the unusual patient group may well prompt special interest.

Reviewer #2 (Remarks to the Author):

The Complementary Role of mPFC and MTL Neurons in Modifying Human Behavior under Goal Conflict

Gazit and colleagues provide the first glimpse into the relationship between the medial temporal lobe

and the medial prefrontal cortex at the single neuron level during adaptation and resolution of goal conflicts. The authors introduce a goal-conflict video game task where the subject receives rewards (coin collection) or punishment (balls) by moving an avatar right or left. This work comes from one of the few groups in the world capable of recording the activity of individual neurons in the human brain, which is one of the particularly exciting aspects of this study. The authors show that the dependence of medial frontal cortex neurons in representing punishment depends on whether subjects can control the outcomes or not. In the hippocampus, the responses to punishment signal were predictive of subsequent adaptive behavior. What is particularly striking is the large differences between the control and no control trials, despite the similarity in the sensory inputs and the rewards, etc. I wonder whether the authors should further emphasize this aspect of the results in the title and/or abstract.

Questions/comments

All the responses seem to be aligned to the specific outcomes (reward, punishment). I assume that subjects could easily predict what the outcome would be several hundred milliseconds before the outcome itself (e.g. the red ball approaching the avatar indicates imminent punishment). Essentially most (if not all) of the neuronal responses seem to occur after the outcome. Is this the case? Were none of the neurons correlated with the outcome expectation before the delivery of the reward or punishment?

I wonder to what extent the responses in the hippocampus can be directly mapped to adaptive behavior in this task versus the need to keep tabs in memory (working memory timescales?) of the task conditions and rewards/punishment. If I understand the video game correctly, in the no-control condition, there does not seem to be any need to keep information in memory, whereas in the control condition, working memory is needed for adaptive behavior. Is this how the authors understand the neuronal responses in the hippocampus? Or is this an alternative description of the findings? In the latter case, why is this explanation less plausible?

The manuscript would benefit from a better description of what is shown where. For example, in Figure 1, there are 4 conditions (reward/punishment, control/no control). The \$ and ball symbols clearly refer to reward and punishment, but how do the colors map onto control or not? The figure legend could also point to b, c, f presumably for the mPFC examples.

Figure 5 appears at the end and I am not sure that it is even cited. This Figure goes a long way towards understanding the task and hence the whole paper. I suggest moving it to Figure 1.

It would be useful to take a quick look at the paper for readability, grammar, etc. Here are a couple of examples

these areas response these areas' response

This sentence in the abstract could use some work: "The described differential involvement of these 45

regions sheds light on the mechanism of known sub-processes in anxiety related psychopathology; the bias towards negative cues and motivational outcomes and the tendency to avoid them when faced with a conflict “

Line 356 seems to start with a reference?

Reviewer #3 (Remarks to the Author):

In Gazit et al., intra-cranial recordings were conducted across multiple brain regions (mPFC and MTL) while eleven epileptic patients underwent a goal-conflict game task (Primo) in which they were presented with high or low goal conflict scenarios, which were resolved by either moving a cartoon avatar on a screen to catch a reward-associated coin while dodging punishment-related balls, or to avoid the punishment balls altogether. Participants were also presented with trials in which the outcomes (reward or punishment) occurred irrespective of their choice action (uncontrolled condition). Their experimental design enabled the assessment of neural activity correlating with outcome valence, control over choice action, and prospective coding. It was found that mPFC neurons encoded negative outcomes more than positive outcomes, but only when participants had control over their action/outcome. MTL (hippocampus and amygdala) neurons, on the other hand, responded to both positive and negative outcomes but also appeared to influence future choice towards avoidance, again only in controlled conditions. The authors concluded that the PFC and MTL subserve differential but complementary processes in encoding controlled punishment, and negatively biasing subsequent choices.

The findings reported in this paper are of potential significance and interest, given the burgeoning interest in the delineation of the neural substrates and circuit mechanisms that underlie decision making under goal conflict. Parsing out the contributions of cortical and medial temporal lobe areas in approach avoidance decision making is an important endeavor. However, I have major reservations concerning the goal-conflict PRIMO task itself, and many aspects of the procedure, data analysis and interpretation, which cast doubt on the main conclusions of the study.

I am not convinced that the goal-conflict game task is inducing any conflict at all, since the task is so heavily approach biased. As reported, subjects in the present study, as well as healthy populations tend to approach 86.6% of 2675 trials, and even in the High Goal Conflict trials, continue to show this approach bias (81.6%). A scenario that induces a 50-50 approach or avoid response would be more convincing as a conflict task. Furthermore, the fact that participants are choosing to approach in the majority of trials means that the approach response is pre-potent in participants, and must be overridden or inhibited in order to produce an avoidance response. Thus, firing rate changes could be related to this inhibition process, rather than the encoding of punishment.

A number of other aspects of the behavioral task lack clarity, and again make me question whether this

task can indeed measure approach vs. avoidance behavior under conditions of conflict. For instance, is it the case that if a participant makes an approach response to a falling virtual coin (e.g. by moving towards the left in Figure 5), they must simultaneously avoid being hit by a falling ball? If so, isn't avoidance behavior inherently part of an approach response? In addition to this, it is not clear what an actual avoid response consists of - is it moving to the right of the screen away from a falling coin? Is it possible to avoid a falling ball by simply staying in the middle of the screen (either remaining still or shifting slightly to the left or the right)? Approach and avoidance responses appear to be conflated in this task. Unless it is absolutely clear that the behavioral task measures what it is purported to measure, interpretation of the neural data is difficult.

Figure 5 appears to depict four possible outcomes for a 'controlled' trial – 1) successful approach leading to reward, 2) unsuccessful approach leading to punishment, 3) successful avoidance leading to reward, 4) unsuccessful avoidance leading to punishment. However, this is not articulated in the text, and no consideration is given to the fact that neural responses may be different for each of these response/outcome combinations. Are mPFC neurons sensitive to punishment regardless of whether an approach or avoidance response was generated? Were data analyses conducted separately for each of the four scenarios? To better appreciate the contribution of the responsive neurons in the different regions, it is critical that the authors indicate how many of the neurons in each region responded positively to 1 vs. 2 vs. 3 vs. 4 conditions.

A significant portion of the theoretical motivation of the study focuses on the role of the hippocampus in approach-avoidance behavior and similarly, elements of the discussion of the reported neural findings are carried out with respect to existing theories/work pertaining to the role of the hippocampus in approach-avoidance behavior. Crucially, however, the main findings from the MTL result from an examination of both hippocampal and amygdala neurons in combination. This is problematic as the authors are effectively attempting to make claims about hippocampal involvement in approach-avoidance behavior while appealing to data that are not specific to the hippocampus.

Although the authors do examine hippocampal and amygdala neurons separately when investigating the correlation between temporal firing following punishment and avoidance in subsequent HGC trials (e.g. page 9), there is, in fact, only a trend towards significance in the hippocampus, which does not justify the claims the authors make.

A key component of the authors' findings is the claim that mPFC neurons responded before MTL neurons, in particular for controlled trials. To provide clear support for this claim, the authors need to provide further details to explain their statement 'This effect was not found for the uncontrolled trials' on page 8. To what extent was this effect not found? Was it the precise demonstrated effect for controlled trials that was not found, or was there no indication at all from exhaustive analyses to suggest that mPFC neurons fired earlier than MTL neurons on controlled trials? For instance, if the authors examined the peaks of firing for mPFC and MTL neurons on uncontrolled trials, are these peaks significantly above baseline and moreover is the temporal distance between these peaks significantly greater than zero?

Please move Figure 5 up as Figure 1.

Reviewer #1 (Remarks to the Author):

A chief merit of this study is the relatively unique electrophysiological recordings afforded by the implanted patient group. This provides a rare window into firing patterns during the win/lose game.

The data patterns themselves are fairly subtle and complex, and do not easily give rise to strong interpretations. The authors do a good job in wrestling with interpretation, and their suggestion that prefrontal negativity bias could reflect an evolutionary importance of avoiding negative outcomes seems plausible. The authors also suggest that the temporal lobe, and hippocampus in particular, update avoidance tendencies after receiving a negative outcome signal from prefrontal cortex. All of this seems reasonable, and is consistent with much literature.

Overall, this is an interesting and unusual study. While the data are not particularly conducive to strong interpretation, the authors make a valiant attempt and the unusual patient group may well prompt special interest.

R: Thank you for the encouragement

Reviewer #2 (Remarks to the Author):

The authors show that the dependence of medial frontal cortex neurons in representing punishment depends on whether subjects can control the outcomes or not. In the hippocampus, the responses to punishment signal were predictive of subsequent adaptive behavior. What is particularly striking is the large differences between the control and no control trials, despite the similarity in the sensory inputs and the rewards, etc. I wonder whether the authors should further emphasize this aspect of the results in the title and/or abstract.

R: We thank the reviewer for this focus related suggestion. There is indeed an overall main effect of control in all regions but only in the mPFC there is interaction with valence (Figure 3). We agree that the controllability is an interesting aspect of our findings, possibly related to the importance of sense of agency and making choices under goal conflict. Although we cannot provide a sure functional explanation for this effect (due to lack of measurement of the participant's experience), following the comment, we now refer to the effect more clearly; **a.** opening paragraph in the intro refer to the importance of adaptive choice in our life (pg. 2), **b.** a paragraph in the discussion highlights theoretical perspectives for the sake of future work (pg. 16-17) and suggest the clinical meaning of controllability in the summary statement (pg. 18-19), **c.** we revised the title to include choice in it which is the main behavioral component the signifies the controlled condition in the game.

Questions/comments

All the responses seem to be aligned to the specific outcomes (reward, punishment). I assume that subjects could easily predict what the outcome would be several hundred milliseconds before the outcome itself (e.g. the red ball approaching the avatar indicates imminent

punishment). Essentially most (if not all) of the neuronal responses seem to occur after the outcome. Is this the case? Were none of the neurons correlated with the outcome expectation before the delivery of the reward or punishment?

R: We thank the reviewer for pointing to this issue. Unfortunately, our paradigm does not allow the evaluation of neural responses at the timing prior to outcome (i.e. anticipation), since this time period is contaminated by movements and simultaneous occurrence of different events (rewards and punishment). When designing the paradigm we cared for an ecological situation that could encourage motivational behavior of approach, paying the price of indistinctiveness of events, prior to outcome. This is now referred to as a limitation (pg. 17).

I wonder to what extent the responses in the hippocampus can be directly mapped to adaptive behavior in this task versus the need to keep tabs in memory (working memory timescales?) of the task conditions and rewards/punishment. If I understand the video game correctly, in the no-control condition, there does not seem to be any need to keep information in memory, whereas in the control condition, working memory is needed for adaptive behavior. Is this how the authors understand the neuronal responses in the hippocampus? Or is this an alternative description of the findings? In the latter case, why is this explanation less plausible?

R: We agree with the reviewer's point of view and in fact, in our discussion we offer several explanations that reflect these suggestions. In specific we presume that the effect of the hippocampus on adaptive behavior in the next trial could be related to its role in an episodic-like memory task (i.e. an arbitrator, pg. 14) or to updating the behavioral decision (through prediction error assessment in the ventral striatum, see pg. 14). Clearly – our design could not differentiate between these options and further research is required as suggested in the discussion.

The manuscript would benefit from a better description of what is shown where. For example, in Figure 1, there are 4 conditions (reward/punishment, control/no control). The \$ and ball symbols clearly refer to reward and punishment, but how do the colors map onto control or not? The figure legend could also point to b, c, f presumably for the mPFC examples.

R: Thank you for noting this clarity issue. We revisited all figures' legends and their references in the texts and made it more concisely tight to each other. Specifically, we have added a color legend to figure 1 (now fig. 2) to clarify the different symbols and indicated the different units.

Figure 5 appears at the end and I am not sure that it is even cited. This Figure goes a long way towards understanding the task and hence the whole paper. I suggest moving it to Figure 1.

R: We agree with this suggestion. This figure is moved to be figure 1. Additionally we modified the graphics in the figure to make it clearer and changed the legend accordingly.

It would be useful to take a quick look at the paper for readability, grammar, etc. Here are a couple of examples

these areas response these areas' response

This sentence in the abstract could use some work: "The described differential involvement of these regions sheds light on the mechanism of known sub-processes in anxiety related

psychopathology; the bias towards negative cues and motivational outcomes and the tendency to avoid them when faced with a conflict “

Line 356 seems to start with a reference?

R: We revised the writing extensively in all parts, and an English-speaker proof-edited the whole manuscript.

Reviewer #3 (Remarks to the Author):

*The findings reported in this paper are of potential significance and interest, given the burgeoning interest in the delineation of the neural substrates and circuit mechanisms that underlie decision making under goal conflict. Parsing out the contributions of cortical and medial temporal lobe areas in approach avoidance decision making is an important endeavor. However, I have major reservations concerning the goal-conflict PRIMO task itself, and many aspects of the procedure, data analysis and interpretation, which cast doubt on the main conclusions of the study. **I am not convinced that the goal-conflict game task is inducing any conflict at all**, since the task is so heavily approach biased. As reported, subjects in the present study, as well as healthy populations tend to approach 86.6% of 2675 trials, and even in the High Goal Conflict trials, continue to show this approach bias (81.6%). A scenario that induces a 50-50 approach or avoid response would be more convincing as a conflict task.*

R: We thank the reviewer for asking us to clarify this critical aspect in our design. While it is true that the task is approach biased, we believe it induces different levels of goal conflict. High and low goal-conflict (HGC, LGC, respectively) conditions in the game scenario were defined for each trial according to the number of potentially punishing balls in the way to reaching the rewarding coin. This operationalization has been comprehensively validated by a previous fMRI study on 50 healthy individuals (Gonen et al. 2016). As expected, there was less approach behavior under HGC than LGC trials (Gonen et al 2016; Figure 3e). Furthermore, brain mapping analysis during approach under HGC vs LGC conditions, showed greater mesolimbic BOLD activity and functional connectivity under HGC trials (Gonen et al 2016; Figure 5 and 6). Lastly, individual differences in approach/avoidance personality tendencies (indicated by standard personality questionnaires) revealed that individuals with approach personality tendency also showed more approach behavior overall and during HGC trials than individuals with avoidance-oriented personality tendency (Gonen et al Figure 3). Intriguingly, among the players in the current study, a similar conflict sensitive behavior was evident (see supp. Figure S1), showing lower approach probability and faster reaction times towards coins in HGC vs LGC trials. We now elaborate on the fMRI study and clarified the operationalization of goal conflict in our task in the intro, methods and discussion. (pg. 5, 18, 19-20, respectively).

*Furthermore, the fact that participants are choosing to approach in the majority of trials means that the approach response is pre-potent in participants, and must be overridden or inhibited in order to produce an avoidance response. Thus, **firing rate changes could be related to this inhibition process, rather than the encoding of punishment.***

R: This is an important comment and we are glad to have the chance to further explain and clarify it in here and in the text. In our study, the firing rate measurements were aligned to the occurrence of the outcome event and not to the appearance of the cue on the screen. It is

important to note that punishment outcomes can come at any time unlinked to a specific behavior. In opposition to reward outcome that is always associated with a choice of behavior, punishment could occur with or without it. We suspect that an inhibition process, should it occur, would take place during the decision-making part, closely after the cue and before a behavioral response is initiated (or inhibited). Thus, neural responses at the time of the outcome should be clean of any residual effects of such inhibition, if occurred. Nevertheless, to make sure that indeed the controlled punishment effect is not related to such differences in processes occurring prior to outcome, we further analyzed the FR following punishment outcomes, and highlight the difference from a rewarding outcome. The possible types of punishment we analyzed were punishment during an unsuccessful approach behavior (while trying to achieve a reward) and punishment obtained without the presence of a reward on the screen (in between reward trials). This analysis revealed that the selectivity to negative outcomes of the mPFC is evident even when taking into account these differences in punishment occurrences (see methods and results sections pg. 9, 20, respectively and sup Fig S5, also attached here). To make sure this response selectivity is unique to controlled trials we show here an additional analysis comparing controlled punishment without reward and during failed approach presenting increased FR over uncontrolled punishment. Both analyses show this effect solely for mPFC neurons.

To note, the new analysis and results in our paper compares positive and negative outcomes under the different controlled conditions. These results show effects across behavioral choices and goal conflict conditions. We only discuss goal-conflict in the part of the paper that looks at the relationship between neural responses and behavior where actual approach behavior accounts.

Figure S5:

Additional analysis figure:

*A number of other aspects of the behavioral task lack clarity, and again make me question whether this task can indeed measure approach vs. avoidance behavior under conditions of conflict. For instance, is it the case that if a participant makes an approach response to a falling virtual coin (e.g. by moving towards the left in Figure 5), they must simultaneously avoid being hit by a falling ball? If so, isn't avoidance behavior inherently part of an approach response. In addition to this, it is not clear what an actual avoid response consists of - is it moving to the right of the screen away from a falling coin? Is it possible to avoid a falling ball by simply staying in the middle of the screen (either remaining still or shifting slightly to the left or the right)? Approach and avoidance responses appear to be conflated in this task. **Unless it is absolutely clear that the behavioral task measures what it is purported to measure, interpretation of the neural data is difficult.***

R: We appreciate this comment, which encouraged us to clarify an important aspect of our paradigm. A trial is classified as high goal-conflict if at the time of the appearance of the coin there are many balls interfering in the avatar's way. Thus, at this moment, choosing to approach the coin is also risking being hit by the balls (i.e. punishment) so there is no conflation of approach and avoidance. This moment is the one relevant for making the decision whether to approach or avoid this specific coin. Indeed, an approach choice could result in later avoiding balls as well, but the decision and behavioral response are not conflated. Avoidance could mean not moving and thus not approaching the coin, or slightly moving away from its direction, but remaining in the middle of the screen without moving might result in balls falling at your direction all the time. In order to classify the behavioral response as approach or no-approach,

we have categorized the trials based on set of features that allow for a computational cutoff, representing the degree of approach (for more details see paper and supplement of Gonen 2016, for convenience critical parts are copied below). In light of this comment we have realized that using the term avoidance to describe less approach might be misleading since we didn't aim to model the avoidance trials or to characterize them neuronally. As our game encourages approach behavior (and indeed leads to more approach overall), we only regard here behaviors as more or less approach (termed here **approach probability**, see Figure 5). [We clarify this terminology issue in the methods (pg. 19) and throughout the paper replaced the word 'avoidance' with 'less approach' or 'low probability to approach']

Copied from Gonen et al., 2016 supplement: Approach\avoidance classification of PRIMO behavioral game events

One of the ecological features of the PRIMO game was that the motivational behavior played in each trial was sequentially chosen by the subject according to his\her individual motivational tendency and specific game-course. Therefore, labeling of game events into approach or avoidance was performed retrospectively in two stages. First, manual classification was performed on all the controlled trials to establish a gold-standard classification. Second, using the manual classification labels we further trained an automatic classifier using a data driven computational model in order to create a standardized classification method with high reliability for current and future uses. Details regarding both classification methods are presented below.

Manual classification: Manual classification was performed by three independent judges. Each judge viewed the games of all participants in a blinded fashion (with no knowledge of subjects' identity, demographic or behavioral data) via a program developed in-house, which indicated the beginning of each Controlled trial. All judges were briefed with the same classification instructions: approach behavior was defined as the trials where subjects moved the figure towards the falling coin; and avoidance behavior was defined as the times subjects moved away from the coin or stayed still in order to avoid a ball hit. Judges were also instructed to classify the trials entirely by explicit behavior and not try to guess subjects' intentions. In case of ambiguity in a specific trial, judges were instructed to tag it as a "mixed" trial. After three independent judges completed the classification of all 220 sessions (55 subjects * 4 sessions each), classifications were compared and each trial was classified according to the behavior assigned by the majority of the judges, resulting in 3*2 Controlled behavioral tags: Approach, Avoidance, Mixed; each under HiGC or LoGC.

Automatic classification: Automatic classification was conducted using a predictive model that was blinded to manual labeling described above. Additionally, since the main objective of the classification model was to efficiently predict the probability of assigning each event to either approach or avoidance, we discarded the events labeled as mixed from our model construction stage. The construction of the model was done in three stages: In the first stage we aggregated the data into a unified database and randomly separated it into three uneven parts: training (60%) cross-validation (20%) and testing (20%). In the second stage, using exploratory analysis, we defined various features (see feature definition section below) that efficiently represented the variable aspects of the controlled events. Then we eliminated classification irrelevant features using Monte-Carlo uninformative feature elimination (MC-UVE) using partial least square linear discriminant analysis (PLS-LDA) ((Li et al., 2014)) and selected the top ten most stable features. Performance assessment was done using both confusion matrices and Receiver Operator Curves (ROC) (see (Stehman, 1997; Florkowski, 2008)). PLS-LDA full feature space model resulted in accuracy of 96.6% and the selected top ten features space PLSLDA model presented accuracy of 95.9%. As can be seen in Figure S1A, using a simplified feature space decreased the amount of false negative cases (i.e. avoidance samples that were misclassified as approach and vice versa) (for full performance confusion matrices see Figure S1Ai-ii). In order to improve the classification process, we further used the same full-feature space with a regularized logistic regression classification Machine Learning Algorithm (MLA), which is known to improve classification while handling over-fitting.

Therefore, in the last stage we trained a regularized logistic regression model on these top 10 features, and measured its performance on both the cross-validation and test dataset parts with 97.8% accuracy in the cross validation set and 97.3% accuracy in the test set (see Figure S1Aiii-iv for complete confusion matrixes and Figure S1B for the comparable ROC curves). Comparison between the cross-validation and the test sets showed that although the MLA model suffered from reduced performance, it was still better than both PLS-LDA models and was well balanced across both false negatives and false positives (see Figure S1.Ai-iii). Therefore we chose the MLA model using the simplified feature space (top-ten features) as the most fitting for this data and used it to label the current dataset.

Top ten feature definitions: Features in the game were roughly separated into two groups: (A) Explicit behavioral features, which were extracted from the keypad input of the participant in response to the unfolding events in the game (see Figure 2A for a graphical representation of one session of the game). Among those, (1) *Response Time (RT)*, (2) *Engagement*, (3) *idle events*, (4) *Idle time*. (5) *Event-Num*. (B) Distance features were extracted from the movement vector of the player and its interaction with the surrounding events (see Figure 2B). For each event we defined the position on screen (width) with respect to the reward location (defined as zero location), effectively separating the screen into two sides so that a negative sign indicated left side presses and vice versa. This allowed us to efficiently define the various distance features: (6) *Distance-from-reward*, (7) *Range*, (8) *Half-Range*. Exploratory analysis revealed that the interaction between the final position of the player and both the middle and start positions were strong predictors of motivational events, therefore, we added (9) *start*end*, and (10) *mid*end* as features. Table S1 details the top ten features and their definitions.

In order to further inspect the relationships between the top ten features and game behavior we randomly sampled 200 approach and 100 avoidance events and applied unsupervised hierarchical classification. Several interesting associations can be extracted from this analysis. First, as expected, large distance from the falling reward was the best predictor of avoidance events, interestingly these events were also associated with longer response times in general and were temporally located in the beginning of the game session. Additionally, in the avoidance events the *Range*, *Half-Range* and *Engagement* were smaller than those of the approach events. Lastly, clear separation between behavioral events was apparent when plotting *Range* vs. *Distance-from-reward* (Figure 2C).

To summarize, using only in game features we were able to accurately label motivational binary behavior (approach/avoidance) using a data driven computational model. This method enables further use of our novel paradigm without resorting to the time consuming and possibly error-prone human labeling.

Figure 5 appears to depict four possible outcomes for a 'controlled' trial – 1) successful approach leading to reward, 2) unsuccessful approach leading to punishment, 3) successful avoidance leading to reward, 4) unsuccessful avoidance leading to punishment. However, this is not articulated in the text, and no consideration is given to the fact that neural responses may be different for each of these response/outcome combinations. Are mPFC neurons sensitive to punishment regardless of whether an approach or avoidance response was generated? Were data analyses conducted separately for each of the four scenarios? To better appreciate the contribution of the responsive neurons in the different regions, it is critical that the authors indicate how many of the neurons in each region responded positively to 1 vs. 2 vs. 3 vs. 4 conditions.

R: Figure 5 (now fig. 1) describes only the behaviors that were classified by our algorithm representing only situations with behavior under goal-conflict. The four situations are in fact: (1) successful approach leading to reward, (2) unsuccessful approach leading to punishment, (3) punishment without a reward cue (unrelated to behavior). (4) Unsuccessful avoidance leading to punishment. Note that situation 4 is rare and therefore not subject of reliable assessment in our data.

Following the reviewer's comment we now further analyzed scenarios 2&3 in order to show the effect of behavior on the negative bias in mPFC. In this new analysis we performed the comparison between reward and punishment twice: Once considering only punishment trials without reward cue and the second time considering only punishment trials during unsuccessful approach behavior (note: rewards are always successful approach trials so there is no need to split reward trials). We found that in both cases, more mPFC neurons responded to punishment over reward and this was not the case for MTL neurons. We thus conclude the negative bias of mPFC neurons, at least in the context of the paradigm described is related to outcome and not sensitive to whether an approach behavior was generated. We clarified these scenarios in the text and added the results of the new analysis in the supplementary materials (pg. 9, 22 and supplementary Figure S5 also shown above)

A significant portion of the theoretical motivation of the study focuses on the role of the hippocampus in approach-avoidance behavior and similarly, elements of the discussion of the reported neural findings are carried out with respect to existing theories/work pertaining to the role of the hippocampus in approach-avoidance behavior. Crucially, however, the main findings from the MTL result from an examination of both hippocampal and amygdala neurons in combination. This is problematic as the authors are effectively attempting to make claims about hippocampal involvement in approach-avoidance behavior while appealing to data that are not specific to the hippocampus.

Although the authors do examine hippocampal and amygdala neurons separately when investigating the correlation between temporal firing following punishment and avoidance in subsequent HGC trials (e.g. page 9), there is, in fact, only a trend towards significance in the hippocampus, which does not justify the claims the authors make.

R: We thank the reviewer for raising this issue, allowing us to improve the statistical grounds for our claim. To increase statistical power we ran the paradigm on three additional patients with implanted electrodes in the hippocampus, amygdala and cingulate cortex. The analysis from these recordings revealed 11 more units in the hippocampus, 10 units in the amygdala and 1

unit in the cingulate. We reanalyzed the data including these neurons. All results and figures are now updated accordingly. In line with our claim, these additional data resulted in significant prediction of the subsequent behavior from hippocampus and not from other areas, as demonstrated by the regression analysis (pg. 10-11)

A key component of the authors' findings is the claim that mPFC neurons responded before MTL neurons, in particular for controlled trials. To provide clear support for this claim, the authors need to provide further details to explain their statement 'This effect was not found for the uncontrolled trials' on page 8. To what extent was this effect not found? Was it the precise demonstrated effect for controlled trials that was not found, or was there no indication at all from exhaustive analyses to suggest that mPFC neurons fired earlier than MTL neurons on controlled trials? For instance, if the authors examined the peaks of firing for mPFC and MTL neurons on uncontrolled trials, are these peaks significantly above baseline and moreover is the temporal distance between these peaks significantly greater than zero?

R: We further explained these analyses and results in the text (pg. 10) and in Figure 5.

Please move Figure 5 up as Figure 1.

R: Thank you for this suggestion. Done and updated numbers

Reviewers' comments:

Reviewer #1 (Remarks to the Author):

This revision builds on the earlier manuscript with new analyses and new discussion of issues, in response to the other reviews. As I wrote of the original version, I think it is difficult to draw strong interpretations from these subtle data patterns, but the authors wrestling well with the interpretation issues, and that is further improved in this revision. I'm not expert in the methods of this type of study, and leave specific data criticisms to the other reviewers.

Overall, in my view, the relatively subtle data patterns and difficulty of drawing strong conclusions might ordinarily make this paper more appropriate for a specialty journal. But the multiple electrode recordings obtained in implanted human patients during the game is potentially exciting. That gives extra interest to this study that may make it suitable for Nat Communications if the editors see fit.

Reviewer #2 (Remarks to the Author):

The authors have done a thorough job in responding to my previous queries. This is a heroic effort to describe the activity of individual neurons in choice related task with and without a sense of agency and will play an important role in shaping our computational modeling of decision making and agency.

I am not sure what the journal rules are but the abstract seems too long to me.

Personally, I feel that the abstract makes too many allusions to mental health disorders. This is mostly speculative, this is not what the manuscript is about.

I would move the speculations about mental health disorders to the discussion and minimize their discussion in the abstract.

Reviewer #3 (Remarks to the Author):

The authors are to be commended for putting in a significant amount of work including the collection of additional data and extra analyses to address my previous comments. The manuscript is improved and on the whole, the authors have done a good job in further clarifying important details of their work and addressing some of my previous concerns. There remain, however, significant concerns that cloud the validity of the authors' claims.

1) Very recent work (e.g., Ramm et al., 2019) has reported that patients with mesial temporal lobe

epilepsy demonstrate poor conflict processing (see also Oerhn et al., 2015). Indeed, in Bach et al., 2014, it was found that patients with mesial temporal lobe epilepsy demonstrated significantly increased approach responses in the face of conflict. Assuming the use of similar types of patients in the present study, there is significant doubt as to whether the present data can reveal the mechanisms that underpin approach-avoidance conflict processing in the healthy brain. Indeed, supportive of this, the proportion of approach responses in the present study is much larger (83.4%) than that which was observed in Gohen et al, 2016, in which healthy participants approached 64% of the time on HGC trials. Moreover, response times on HGC trials in the present study were shorter than those for LGC trials, which is contrary to what is typically reported in the literature. What accounts for these differences? It is critical that the authors reveal what type of epileptic patients they have studied, and whether those with seizures emanating from the MTL exhibit distinct profiles of behavioral responding and neural activity.

2) One very significant concern has arisen following the authors' clarification of how the neural data were analysed. The fact that the analysis of the electrophysiological data was conducted in relation to the outcome of each trial, rather than the onset of the cue stimulus, is a critical issue and in my opinion, completely changes the theoretical complexion of the study. More specifically, this approach does not provide insight into the role of MTL/mPFC neurons in decision making under conflict per se but rather, post-outcome processing and the impact of feedback on future/subsequent decision making. As an illustration, fMRI studies of conflict processing that have implicated the hippocampus such as Bach et al. (2014), O'Neil et al. (2015), and Loh et al. (2016) have typically examined BOLD response in relation to cue onset rather than trial outcome, thereby providing insight into the neural activity associated with the processes leading up to the moment when an approach-avoidance conflict decision is made. Indeed, in O'Neil et al. (2015), it appears that trial outcome was withheld from participants, thereby removing any possible confounds associated with trial outcome. What the authors are currently reporting is very interesting, but it does not fit within the theoretical framework in which the data are being discussed, and this distinction needs to be made. More importantly, the authors clearly have the data to investigate neural firing in relation to cue onset, and the logical next question, therefore, is whether they have examined their data from this perspective. It seems important that analyses pertaining to both cue onset and outcome onset are reported, and without the former, the theoretical framing and discussion of the data needs considerable revision.

Other concerns:

I appreciate why the authors have focused in on the HGC trials, but I am curious to know what the profile of neural activity was during the controlled LGC reward trials. If, as the authors claim, the reported HGC findings are specific to high conflict, then one would expect there to be significant differences in neural activity between controlled HGC reward trials and controlled LGC reward trials.

It needs to be made clear in both the results and figure legends that time 0 indicates the onset of trial outcome.

Response to Reviewer's comments

We thank the reviewers for the careful consideration of our manuscript and revision. Below is a point-to-point response to the comments raised by the reviewers. Changes to the manuscript are colored in **blue** font within the manuscript document.

Reviewer #1 (Remarks to the Author):

This revision builds on the earlier manuscript with new analyses and new discussion of issues, in response to the other reviews. As I wrote of the original version, I think it is difficult to draw strong interpretations from these subtle data patterns, but the authors wrestling well with the interpretation issues, and that is further improved in this revision. I'm not expert in the methods of this type of study, and leave specific data criticisms to the other reviewers.

Overall, in my view, the relatively subtle data patterns and difficulty of drawing strong conclusions might ordinarily make this paper more appropriate for a specialty journal. But the multiple electrode recordings obtained in implanted human patients during the game is potentially exciting. That gives extra interest to this study that may make it suitable for Nat Communications if the editors see fit.

Reviewer #2 (Remarks to the Author):

The authors have done a thorough job in responding to my previous queries.

This is a heroic effort to describe the activity of individual neurons in choice related task with and without a sense of agency and will play an important role in shaping our computational modeling of decision making and agency.

I am not sure what the journal rules are but the abstract seems too long to me.

Comment: Personally, I feel that the abstract makes too many allusions to mental health

disorders. This is mostly speculative, this is not what the manuscript is about.

I would move the speculations about mental health disorders to the discussion and minimize their discussion in the abstract.

R: We thank the reviewer for this comment. We shortened the abstract by removing some of the discussion in relation to mental health.

Reviewer #3 (Remarks to the Author):

The authors are to be commended for putting in a significant amount of work including the collection of additional data and extra analyses to address my previous comments. The manuscript is improved and on the whole, the authors have done a good job in further clarifying important details of their work and addressing some of my previous concerns. There remain, however, significant concerns that cloud the validity of the authors' claims.

Comment: 1) Very recent work (e.g., Ramm et al., 2019) has reported that patients with mesial temporal lobe epilepsy demonstrate poor conflict processing (see also Oerhn et al., 2015). Indeed, in Bach et al., 2014, it was found that patients with mesial temporal lobe epilepsy demonstrated significantly increased approach responses in the face of conflict. Assuming the use of similar types of patients in the present study, there is significant doubt as to whether the present data can reveal the mechanisms that underpin approach-avoidance conflict processing in the healthy brain. Indeed, supportive of this, the proportion of approach responses in the present study is much larger (83.4%) than that which was observed in Gonen et al, 2016, in which healthy participants approached 64% of the time on HGC trials. ...What accounts for these differences? It is critical that the authors reveal what type of epileptic patients they have studied, and whether those with seizures emanating from the MTL exhibit distinct profiles of behavioral responding and neural activity.

R: We thank the reviewer for this comment. Indeed, this study, as all studies using depth electrode recordings from patients suffer from this limitation. In our discussion on page 18 we refer to this limitation:

Yet, since our data was obtained from patients with epilepsy, the generalization of these results to other populations should be considered with caution.

Specifically, as stated by the reviewer, there have been reports of reduced avoidance (increased approach) in patients with MTL lesions. In accordance with the reviewer's suggestion we added a clinical table describing SOZ localization (according to the iEEG study) and surgical outcome when available (page 25). Five of our 14 patients had a seizure onset zone within the MTL and one within the mPFC (although there were no neurons recorded within this mPFC area). The remaining 8 patients had SOZ not within the micro wire recording sites (in more lateral sites which only had macro wires, or in areas not monitored by the invasive implantation - see table).

To further examine the possibility that MTL patients exhibited different behavioral and neural tendencies, we've added several measures:

1. We evaluated whether there was a difference in approach tendencies between patients with MTL lesions (5) vs. patients with other cortical or subcortical lesions (9). There was no difference in approach-avoidance behavior between these groups. We added the following statement to the results section (page 6-7):

Approach tendencies did not differ between patients with an MTL Seizure Onset Zone (SOZ) (5 patients, 84.7% and 91.1% approach for HGC and LGC respectively) and patients with extra-MTL SOZ (9 patients, 80.8% and 92.7% approach for HGC and LGC respectively). [Mann-Whitney test, $U=19.5$, $Z=0.61$, $p>0.05$ for HGC and $U=19$, $Z=0.07$, $p>0.05$].

2. In accordance with the reviewers' comment, we checked approach tendencies of a different population with the PRIMO task. The task was performed in a quiet laboratory room by 20 healthy volunteers who, like our patients, did not receive monetary reward for their performance in the task. As expected, approach probability in this group was higher for the LGC condition [92.4% (± 0.05)] compared to the HGC condition [78% (± 0.14)] [$t(19)=6.02$,

$p < 0.00001$]. Approach probabilities are relatively similar to those of our group and there was no statistical difference between this healthy population and our patients for the high [$t(33)=0.94$, $p=0.35$] or low [$t(33)=0.32$, $p=0.75$] goal conflict conditions. These results were added to the supplementary materials.

The difference in approach probability between our study group and that of the original paper (Gonen et al., 2016) may be a result of the different setting in which subjects performed the task. In the original paper participants received monetary compensation, which they were told was related to their achievements in the task. This may have resulted in more conservative behavior patterns; i.e. less approach behavior in order to avoid loss of real money. Moreover, in the original paper, participants performed the task within the MRI scanner, a stressful scenario to some participants (the effect of stress on approach avoidance has long been established [Haller and Bakos, 2002]). It may be difficult to draw conclusions from comparing the specific behavioral values of approach tendencies and RTs between these studies.

3. We evaluated whether the observed effect of MTL firing following a Punishment outcome on subsequent HGC trials was affected by neurons within the MTL SOZ. We found it was not and added the following to the results section (page 11):

Even when removing MTL neurons (2 responsive neurons from the left amygdala of patient 6) that were within the SOZ, this finding remained significant [$\beta=1.2$, $t=4.3$, $p < 0.0001$, FDR corrected].

4. We evaluated whether neurons within the SOZ had an effect on the probability to respond to the different Outcome conditions, or on the stronger response of mPFC vs MTL neurons for Control Punishment Outcomes over Uncontrolled Punishment and Control Reward. These analyses found that the same results are replicated even when removing neurons within the SOZ. We now summarize this as a section of the supplementary material and refer to this in the results section:

Page 8: ...even after removing neurons within the MTL SOZ (supplementary materials)

Page 9: ...The results were still significant when removing neurons within the SOZ from the analysis

Thus, given these results we have reason to believe our findings are not specifically a result of MTL lesions or epilepsy.

Nevertheless, we added a discussion of this limitation on page 18:

...since our data was obtained from patients with epilepsy the generalization of these results to other populations should be considered with caution. This may be particularly relevant, since it has been previously suggested that patients with temporal lobe lesions present more approach (or less avoidance) behavior compared to controls³⁰. In addition, studies using Stroop related paradigms showed that hippocampal activation is important for conflict resolution⁵⁰ and that patients with MTL lesions present impaired performance on these conflict tasks^{51,52}. Other studies however found no major difference in the Stroop task between MTL patients and healthy controls⁵³. We believe our findings are not specific to MTL lesions or epilepsy for several reasons. First, only five of the 14 patients had seizures originating from the MTL, and these five patients did not exhibit different approach tendencies compared to the extra-MTL patient group. Second, removing the few neurons from within the epileptic SOZ in the MTL did not change the significance of the results, either neuronally or behaviorally. Lastly, approach probabilities and reaction times obtained from our group of patients were similar to those obtained from a control group of 20 healthy participants (see supplementary Material).

Comment: Moreover, response times on HGC trials in the present study were shorter than those for LGC trials, which is contrary to what is typically reported in the literature.

R: To examine if the RT results are related to our patient group we compared our reaction times to those which were measured in the previously described healthy group and found that, similar to our patient cohort, healthy subjects had shorter RTs for the HGC trials (780.6 ± 115.7) vs LGC trials (849.4 ± 120.6) [$t(19)=3.64$, $p<0.002$]. It is our opinion that shorter reaction times during HGC trials may be a result of task related demands, as a faster response is necessary to avoid punishment. This explanation was added to the discussion, page 6:

Shorter reaction time during the HGC condition may be a result of task related demands, as a faster response is necessary to avoid punishment when facing multiple threats.

Similar findings have been reported by others. For example, using a facial expression approach-avoidance task, Heuer et al., showed that participants exhibited faster responses to emotional faces compared with neutral faces [Heuer et al., 2007]. The approach avoidance scenario is different from that of the classic STROOP conflict tasks, in which there is no negative consequence for the delay in participant reaction and no difference in this aspect between the conflicting and the non-conflicting scenarios.

Comment: 2) One very significant concern has arisen following the authors' clarification of how the neural data were analyzed. The fact that the analysis of the electrophysiological data was conducted in relation to the outcome of each trial, rather than the onset of the cue stimulus, is a critical issue and in my opinion, completely changes the theoretical complexion of the study. More specifically, this approach does not provide insight into the role of MTL/mPFC neurons in decision making under conflict per se but rather, post-outcome processing and the impact of feedback on future/subsequent decision making. As an illustration, fMRI studies of conflict processing that have implicated the hippocampus such as Bach et al. (2014), O'Neil et al. (2015), and Loh et al. (2016) have typically examined BOLD response in relation to cue onset rather than trial outcome, thereby providing insight into the neural activity associated with the processes leading up to the moment when an approach-avoidance conflict decision is made. Indeed, in O'Neil et al. (2015), it appears that trial outcome was withheld from participants, thereby removing any possible confounds associated with trial outcome. What the authors are currently reporting is very interesting, but it does not fit within the theoretical framework in which the data are being discussed, and this distinction needs to be made. More importantly, the authors clearly have the data to investigate neural firing in relation to cue onset, and the logical next question, therefore, is whether they have examined their data from this perspective.

R: We appreciate the reviewers' important comment. We now further stress that the analysis approach is focused on the outcome period throughout the introduction, discussion and methods sections (see below). Our decision to focus on the response to outcome period is

related to our paradigm, designed to be balanced for outcome trials between conditions (i.e. Controlled/Uncontrolled - Reward/Punishment) rather than being balanced for the cues' appearance. The task was designed as such because we specifically aimed to examine the relations between response to outcome and subsequent behavior. This rationale, is now more clearly indicated in the introduction, including pointing to its relevance to known abnormalities in motivation behavior among psychiatric patients [Eshel and Roiser et al., 2010].

We added the following to the introduction:

Page 3: The current study aims to unravel the neural process that underlies response to outcomes of such goal-conflicts and its effect on subsequent behavioral choices.

Page 4: patients suffering from depression are unable to exploit affective information to guide behavior (Eshel and Roiser, 2010).

Page 5: Yet, it remains to be seen whether these results, pointing to the significance of the MTL in the processing of outcomes and adapting behavior accordingly, are relevant to outcomes which appear in the context of an approach-avoidance conflict.

Page 5: To investigate the neural circuitry of reward and punishment outcomes and their effect on decision making under an approach-avoidance conflict, we used...

The following was added to the methods:

Page 22: As this study concentrates on neural response to outcome, time 0 relates to the moment when the ball or coin hit the avatar (time of outcome), throughout the manuscript.

Importantly, following the theoretical concern we changed the introduction to review the different possible roles of the MTL, and specifically hippocampal, during the decision itself vs. during outcome processing affecting future decision. Indeed, in classical approach-avoidance paradigms this distinction is difficult to assess and we further present recent findings and theories relating to this.

The following has been added/modified to update the theoretical framing:

Pages 3-5: The current study aims to unravel the neural process that underlies response to outcomes of such goal-conflicts and its effect on subsequent behavioral choices.

Classical animal studies using goal-conflict paradigms such as the Elevated Plus Maze (EPM)¹¹ and the discrete trial runway approach-avoidance tasks¹² have implicated the amygdala¹³, hippocampus^{9,14} and mPFC^{15,16} as being crucial in triggering avoidance behavior in goal conflict situations. For example, in the classic work of Kimura et al.,¹² rats were punished with a delivery of an electrical shock as they consumed food (avoidance training). Over time, control animals increased their latency to enter the target box while rats with hippocampal lesions presented impaired acquisition of such passive avoidance behavior. However, classical animal model studies have not clearly differentiated the neural substrates involved in using information regarding the valence of outcomes (reward versus punishment) for subsequent adaptation of approach behavior, from those that mediate the actual resolution of the goal conflict when potential rewards and punishments are simultaneously present¹⁷. Schumacher et al.,¹⁷ attempted to dissociate between these processes in rodents using lesions to the ventral hippocampus in a mixed valence conditioning paradigm. Their results show that the ventral hippocampus is involved in the resolution of approach-avoidance conflict at the moment of decision making rather than in learning about the value of outcomes for future decisions. On the other hand, further studies in humans and animals showed that the hippocampus, as well as the amygdala, seem to support learning from outcomes and thus affect future behavior. For example, using an fMRI reinforcement learning task, Davidow et al.⁵ showed that adolescents were better than adults at learning from outcomes to adapt subsequent decisions, and that this was related to heightened prediction error-related BOLD activity in the hippocampus. Using lesions to macaque amygdalae, Costa et al.,¹⁸ present evidence that the amygdala plays an important role in learning from outcomes to influence subsequent choice behavior. With relation to psychopathology, it has been suggested that patients suffering from depression are unable to exploit affective information to guide behavior¹⁹. For example, using an event related fMRI task in which thirsty subjects had to associate images with reward (water) probability, Kumar et al.,²⁰ found reduced reward learning signals (learning from outcomes to update behavior) in the hippocampus and

anterior cingulate in patients suffering from major depression. Disruption of prediction-outcome associations in the bilateral amygdala–hippocampal complex was found in patients with schizophrenia, with the degree of disruption correlating with psychotic symptom severity²¹. Yet, it remains to be seen whether these results, pointing to the significance of the MTL in the processing of outcomes and adapting behavior accordingly, are relevant to outcomes which appear in the context of an approach-avoidance conflict.

We also removed some of the background which was more relevant for the approach-avoidance decision phase.

Comment: It seems important that analyses pertaining to both cue onset and outcome onset are reported, and without the former, the theoretical framing and discussion of the data needs considerable revision.

R: In accordance with the reviewer’s comment, we now add an analysis of the cue period (see supplementary materials). Please note however, that since our paradigm is controlled for outcomes, the part of the cue is not balanced as it was mainly intended to induce an engaging and conflicting game environment; i.e. not all coins in the Controlled condition were caught and not all balls hit the avatar. As a result, there were more cue appearances than outcomes for the Controlled condition with more ball appearances than coins. This is not the case for the Uncontrolled condition, in which all cues were caught, regardless of behavior. This difference in trial number makes the interpretation of these results more complicated, as saliency effects may mask condition differences.

This methodological issue is discussed as a limitation of our design (page 15):

Unfortunately, our design did not allow an objective evaluation of neural response directly following cue appearance due to unbalanced trials across the different conditions (see supplementary material), resulting in confounding saliency effects. Future studies with a similar design but balanced cue trials are warranted to evaluate the MTL’s role during the decision making phase.

The following was added as supplementary:

Neural response to cue: The number of cues is different between conditions. To account for this, we randomly selected some of the Controlled trials to match the amount of the Uncontrolled trials, resulting in an equal amount of trials in all conditions. While this accounts for the statistical biases it does not account for the cognitive bias as participants comprehend Uncontrolled conditions as more rare and thus a saliency effect may manifest in the results. We found 23 of 79 (29%), 27 of 61 (44%), 20 of 63 (32%) and 40 of 107 (37%) neurons that significantly responded to at least one of the four cue conditions (Controlled/Uncontrolled Reward/Punishment) in the Amygdala, Hippocampus, dmPFC and CC respectively.

When examining response probability to Controlled and Uncontrolled cues across valence type (Rewards and Punishments), a higher probability to respond to the Uncontrolled condition over Controlled condition was apparent in neurons from all four areas, but was significant only in the amygdala [$p=0.01$ FDR corrected McNamer test]. Also, a higher probability to respond to the Reward over Punishment condition was apparent only in the amygdala [$p=0.005$ FDR corrected McNamer test]. Unlike in the analysis of outcomes, response selectivity to the valence of the cues under the Controlled condition did not differ between regions [$\chi^2=0.46$, $p=0.93$]. Response selectivity to the valence of the cues under the Uncontrolled condition also did not differ between regions [$\chi^2=2.1$, $p=0.56$].

We interpret these results in the context of saliency. Interestingly, both the controllability and valence effects are strongest in the amygdala, a major hub of the brain's salience network [Seeley et al., 2007].

Along with the reviewers comment, we update the theoretical framing of these results. The changes to the introduction were described in the previous answer.

The following was added to the discussion (pages 14-15):

However, diverging from the RST model, our results point to the significance of the MTL, not only in the online processing of positive and negative reinforcements but also in the use of such information to influence future motivation behavior. This fits well with the hippocampus' known role in association learning and extinction. For example Davidson and Jarrard³² extend the RST model by proposing that the hippocampus is needed to form inhibitory associations

between events that are concurrently embedded in excitatory associations. Thus, in our study, there is a broad association between rewarding cue and positive outcome. When a participant encounters a punishment concordant with a rewarding cue it must inhibit this association, and this may be performed through hippocampal signaling. The formation of this inhibitory association could be the critical process by which the hippocampus increases the weight of affectively negative information to influence decision.

Other concerns:

Comment: I appreciate why the authors have focused in on the HGC trials, but I am curious to know what the profile of neural activity was during the controlled LGC reward trials. If, as the authors claim, the reported HGC findings are specific to high conflict, then one would expect there to be significant differences in neural activity between controlled HGC reward trials and controlled LGC reward trials.

R: As we were interested in the effect of outcome on behavior, we evaluated neural responses to outcome and their relation to subsequent decision, and not neural response to cue or during the choice of behavior while facing the conflict. We found that neural response to Punishment outcomes in the MTL correlated with future decision during HGC conditions. Since avoidance in LGC trials was rare we did not concentrate on predicting behavior in these trials. In line with the reviewers' comment we computed a generalized linear mixed model to predict behavior in subsequent LGC trials. In this analysis, firing following Punishment outcome did not predict behavior in subsequent LGC trials, showing that this effect is specific for predicting behavior under HGC. We add this analysis to results section:

Page 11: ... This result was not replicated for the LGC trials; MTL response to punishment did not predict subsequent behavior under LGC.

Comment: It needs to be made clear in both the results and figure legends that time 0 indicates the onset of trial outcome.

R: Thank you for noting this, we changed the figures and legends accordingly

References:

Eshel, Neir, and Jonathan P. Roiser. "Reward and punishment processing in depression." *Biological psychiatry* 68.2 (2010): 118-124.

Haller, J., and N. Bakos. "Stress-induced social avoidance: a new model of stress-induced anxiety?." *Physiology & behavior* 77.2-3 (2002): 327-332.

Heuer, Kathrin, Mike Rinck, and Eni S. Becker. "Avoidance of emotional facial expressions in social anxiety: The approach–avoidance task." *Behaviour research and therapy* 45.12 (2007): 2990-3001.

Hill, Michael R., Erie D. Boorman, and Itzhak Fried. "Observational learning computations in neurons of the human anterior cingulate cortex." *Nature communications* 7.1 (2016): 1-12.

Hollerman, Jeffrey R., Leon Tremblay, and Wolfram Schultz. "Influence of reward expectation on behavior-related neuronal activity in primate striatum." *Journal of neurophysiology* 80.2 (1998): 947-963.

Paton, Joseph J., et al. "The primate amygdala represents the positive and negative value of visual stimuli during learning." *Nature* 439.7078 (2006): 865-870.

***REVIEWERS' COMMENTS:

Reviewer #3 (Remarks to the Author):

The authors have done a significant amount of work to address our additional comments. In particular, reframing the paper with regards to examining neural activity in relation to trial outcome, and further clarifying the potential impact of location of SOZ location have been extremely helpful. Similar to Reviewer 1, I still feel that the nature of the paradigm and data make it difficult to draw strong conclusions from the data. Nevertheless, the manuscript is now substantially improved compared to the original version.